# PTGS is dispensable for the initiation of epigenetic silencing of an active transposon in *Arabidopsis*

Marieke Trasser [ID][1,2,3], Grégoire Bohl-Viallefond [ID][1], Verónica Barragán-Borrero [ID][1], Laura Diezma-Navas[1], Lukas Loncsek[1], Magnus Nordborg [ID][1] & Arturo Marí-Ordóñez [ID][1✉]

## Abstract

**Transposable elements (TEs) are repressed in plants through transcriptional gene silencing (TGS), maintained epigenetic silencing marks such as DNA methylation. However, the mechanisms by which silencing is first installed remain poorly understood in plants. Small interfering (si)RNAs and post-transcriptional gene silencing (PTGS) are believed to mediate the initiation of TGS by guiding the first deposition of DNA methylation. To determine how this silencing installation works, we took advantage of *ÉVADÉ (EVD)*, an endogenous retroelement in *Arabidopsis*, able to recapitulate true de novo silencing with a sequence of PTGS followed by a TGS. To test whether PTGS is required for TGS, we introduce active *EVD* into RNA-DEPENDENT-RNA-POLYMERASE-6 (RDR6) mutants, an essential PTGS component. *EVD* activity and silencing are monitored across several generations. In the absence of PTGS, silencing of *EVD* is still achieved through installation of RNA-directed DNA methylation (RdDM). Our study shows that PTGS is dispensable for de novo *EVD* silencing. Although we cannot rule out that PTGS might facilitate TGS, or control TE activity, initiation of epigenetic silencing can take place in its absence.**

**Keywords** Silencing; Transposons; siRNAs; Epigenetics; Plants
**Subject Categories** Chromatin, Transcription & Genomics; Plant Biology

## Introduction

Due to their mobile nature, transposable elements (TEs) pose a threat to genome integrity and can cause mutations compromising host fitness (McClintock, 1984; Bennetzen and Wang, 2014; Schubert and Vu, 2016; Bourque et al, 2018). TEs are mostly transcriptionally repressed across genomes through the action of epigenetic silencing mechanisms, limiting TE activity (Lippman et al, 2004; Allshire and Madhani, 2018).

In *Arabidopsis thaliana*, the best studied model for plants, DNA methylation (5-methylcytosine; 5mC) and histone H3 lysine-9 di-methylation (H3K9me2) cooperatively mediate transcriptional gene silencing (TGS) of TEs (Bernatavichute et al, 2008). Once established, 5mC and H3K9me2 patterns are propagated across generations to ensure transgenerational silencing of TEs. Maintenance of cytosine methylation depends on its sequence context, CG, CHG, or CHH (where H can be any nucleotide besides G). METHYLTRANSFERASE 1 (MET1) preserves 5mC in the CG context after each DNA replication cycle (Mathieu et al, 2007). Maintenance in CHG and CHH contexts occurs through a self-reinforcing loop with H3K9me2 (Chan et al, 2005; Du et al, 2012; Law et al, 2013; Du et al, 2014). In addition, DNA methylation deposition, particularly in the CHH context, can be mediated by the RNA-directed DNA methylation (RdDM) pathway. Canonical RdDM relies on the action of two plant specific RNA polymerases, PolIV and PolV. On the one hand, PolIV is recruited to TE loci through the histone mark H3K9me2. PolIV transcripts are converted into dsRNAs. These are further processed into 24-nt siRNA by DICER-LIKE 3 (DCL3), which are loaded into ARGONAUTE (AGO) 4/6-clade proteins. On the other hand, PolV is recruited to DNA-methylated TE loci. PolV provides the scaffold/target transcript for loaded AGO proteins, guiding the deposition of DNA methylation in all cytosine contexts (Law and Jacobsen, 2010; Kuhlmann and Mette, 2012; Matzke and Mosher, 2014; Erdmann and Picard, 2020). Given the dependency on PolIV-derived siRNAs, this branch of RdDM is also known as PolIV-RdDM.

While the maintenance of TE silencing relying on pre-existing heterochromatic marks is well described, the deposition of de novo silencing marks on active, proliferative transposable elements remains poorly understood. To gain insight into the initial molecular events by which plants recognize and silence active TEs, several studies have investigated host responses to environmentally, chemically, developmentally, or genetically induced TE reactivation (Teixeira et al, 2009; Slotkin et al, 2009; Mirouze et al, 2009; Reinders et al, 2009; Ito et al, 2011; Thieme et al, 2017). One of the best studied TEs in *Arabidopsis* is the *Ty1/Copia* long terminal repeat (LTR) retrotransposon *ÉVADÉ* (*EVD*; *Copia93*) (Mirouze et al, 2009). *EVD* is a functional, low copy TE in the reference Col-0 *Arabidopsis* ecotype and mainly regulated through CG methylation. Therefore, it can be released from silencing through loss of MET1 or the chromatin remodeler DECREASE IN

[1]Gregor Mendel Institute of Molecular Plant Biology (GMI) of the Austrian Academy of Sciences, Vienna 1030, Austria. [2]Vienna BioCenter PhD Program, Doctoral School of the University of Vienna and Medical University of Vienna, Vienna, Austria. [3]Present address: Howard Hughes Medical Institute, Cold Spring Harbor Laboratory, 1 Bungtown Rd, Cold Spring Harbor, NY 11724, USA. ✉E-mail: arturo.mari-ordonez@gmi.oeaw.ac.at

DNA METHYLATION 1 (DDM1). Once lost, CG methylation cannot be reestablished, thus reactivated *EVD* remains active even after the reintroduction of functional (wild type) alleles of either *MET1* or *DDM1* and can rapidly increase in copy number all over the genome (Mathieu et al, 2007; Reinders et al, 2009; Mirouze et al, 2009). Owing to such property, *EVD* has quickly become a model system to study retrotransposon biology, TE bursts, and de novo silencing phenomena (Mirouze et al, 2009; Tsukahara et al, 2009; Marí-Ordóñez et al, 2013; Oberlin et al, 2017).

The *EVD* genome colonization and silencing cycle can be divided in well-defined stages. First, upon *EVD* reactivation, post-transcriptional gene silencing (PTGS) acts as initial host response (Fig. 1A) (Marí-Ordóñez et al, 2013; Oberlin et al, 2022). *EVD* PTGS is the result of its transcriptional and translational strategy to complete its transposition cycle. Due to an alternative splicing event, *EVD* produces two transcripts: (i) a full-length polycistronic mRNA (*flGAG-POL*) encoding for both its structural (Gag nucleocapsid) and catalytic (Pol) components; (ii) a short, Gag only transcript (*shGAG*), which is preferentially translated to generate the molar excess of Gag-to-Pol needed for the formation of virus-like particle (VLP) (Oberlin et al, 2017; 2022). A ribosome stalling event triggered during *shGAG* translation leads to the cleavage of the transcript. The resulting 3′ RNA fragment becomes a substrate for the RNA-DEPENDENT RNA POLYMERASE 6 (RDR6) to produce double-stranded (ds)RNA, further processed by DICER-LIKE 4 (DCL4) into *GAG*-derived 21-nt siRNA (Oberlin et al, 2022). Albeit PTGS does reduce *EVD GAG* mRNA and protein levels, it does not prevent *EVD* transposition (Marí-Ordóñez et al, 2013; Oberlin et al, 2022). Next, *EVD* continues to increase its copy number across generations. When a threshold of 40–50 *EVD* copies per genome is reached, the excess of dsRNA produced by RDR6 is eventually processed by DCL3, giving rise to a population of *GAG*-derived 24-nt siRNAs (Fig. 1A). Loaded into AGO4-clade proteins, they guide the deposition of DNA methylation at *EVD-GAG*-coding sequences in a non-canonical RdDM pathway also known as RDR6-RdDM (Nuthikattu et al, 2013; Marí-Ordóñez et al, 2013). Finally, following *GAG* methylation, a switch from PTGS to transcriptional gene silencing (TGS) takes place. *EVD* TGS is characterized by the installation of PolIV-RdDM and the production of 24-nt siRNAs from *EVD-LTR* sequences, mediating DNA methylation of its regulatory sequences and the de novo TGS of new *EVD* copies (Fig. 1A) (Marí-Ordóñez et al, 2013).

These observations have led to a model under which RDR6-RdDM is thought to contribute to the switch from PTGS to TGS by initiating the deposition of DNA methylation on *EVD GAG*-coding sequences. PTGS-derived siRNAs have been shown to play a role in restoring DNA methylation patterns in DNA methylation mutants (Teixeira et al, 2009; Nuthikattu et al, 2013; McCue et al, 2015) or during key developmental processes involving epigenetic reprogramming (Slotkin et al, 2009). Hence, RDR6-RdDM has been suggested as essential step to gap the transition from PTGS to TGS in the process of de novo transposon silencing (Marí-Ordóñez et al, 2013; Nuthikattu et al, 2013; McCue et al, 2015; Panda et al, 2016). More recently, using an *EVD* transgenic system, a mechanism for such transition has been proposed, under which AGO4, loaded with RDR6-derived siRNAs, interacts with PolII *EVD* transcripts at *EVD* loci to recruit PolV and downstream silencing components to ignite self-sustained PolIV-RdDM (Sigman et al, 2021). However, the requirement of PTGS for the initiation of TGS during a TE

colonization event has never been experimentally tested. Furthermore, upon genome-wide loss of DNA methylation, the majority of reactivated TEs triggering RDR6-dependent PTGS are decayed TE-remnants incapable of transposition, while those intact enough to engage in translation do not (Oberlin et al, 2022). This has casted doubt on the universality of PTGS as a sensor of active TEs and initiator of epigenetic silencing during a TE colonization event.

In this study we address the necessity of PTGS and RDR6-RdDM in the process of de novo silencing of transposable elements. To achieve this, active *EVD* was introduced into *RDR6* mutant plants to prevent RDR6-RdDM. *EVD* activity and silencing status were monitored over several generations in wild-type (WT) and *rdr6* lines. While the mutation of *RDR6* successfully prevented the production of siRNAs involved in RDR6-RdDM, silencing of *EVD* through installation of PolIV-RdDM in its LTRs was achieved in both, WT and mutant background. Therefore, PTGS is not essential for the installation of epigenetic silencing at active *EVD* copies. Given that *EVD* presents a unique case in triggering PTGS upon reactivation in the first place, we suggest that PTGS and RDR6-derived siRNA play a role in limiting *EVD* expression and transposition rather than initiating de novo silencing.

# Results

## Absence of RDR6 does not prevent the production of *EVD-LTR* 24-nt siRNAs

To investigate the role of PTGS and RDR6-RdDM in the initiation of canonical RdDM on *EVD*, and given that RDR6 is responsible for the production of *EVD* siRNA during the PTGS phase (Oberlin et al, 2017; 2022), *rdr6-15* mutant plants were crossed to the 8th generation of the *met1*-derived epigenetic recombinant inbred line (epiRILs) number 15 (epi15 F8), a generation in which RDR6-RdDM has not yet been activated (Fig. 1B) (Marí-Ordóñez et al, 2013; Oberlin et al, 2022). This minimized the probability of introducing active *EVD* copies with *GAG* DNA methylation in the cross to *rdr6*. In the second generation (F2), two homozygous RDR6 wild-type (RDR6) and two mutant (*rdr6*) plants were selected by genotyping (hereafter referred to as RDR6-*EVD* and *rdr6-EVD*, respectively). They were allowed to self-pollinate in order to bulk-propagate two independent WT and two independent mutant populations until the 6th generation (F6), allowing *EVD* to colonize the genome (Fig. 1B). The silencing status of *EVD* in the two independent respective populations was monitored in bulks of 8 plants at generations F2, F4, and F6 by assessing the *EVD* copy number and expression levels through qPCR as well as the siRNA profile by RNA blot.

*EVD* copy numbers consistently increased across generations, confirming the inheritance of active, transposition-competent *EVD* copies from epi15 (Fig. 1C). *EVD* copies accumulated at a higher rate in *rdr6-EVD* background than in *RDR6-EVD*. The estimated copy number was consistently higher in *rdr6-EVD* than in *RDR6-EVD* plants, and the difference was significant in the F6, where loss of RDR6 activity allowed the accumulation of over 100 *EVD* copies (Fig. 1C). Consistent with an increasing copy number, *EVD* transcripts were also more abundant up to the 4th generation, particularly in *rdr6-EVD*. However, at the F6, large variations between biological replicates were observed in both *RDR6-EVD* and

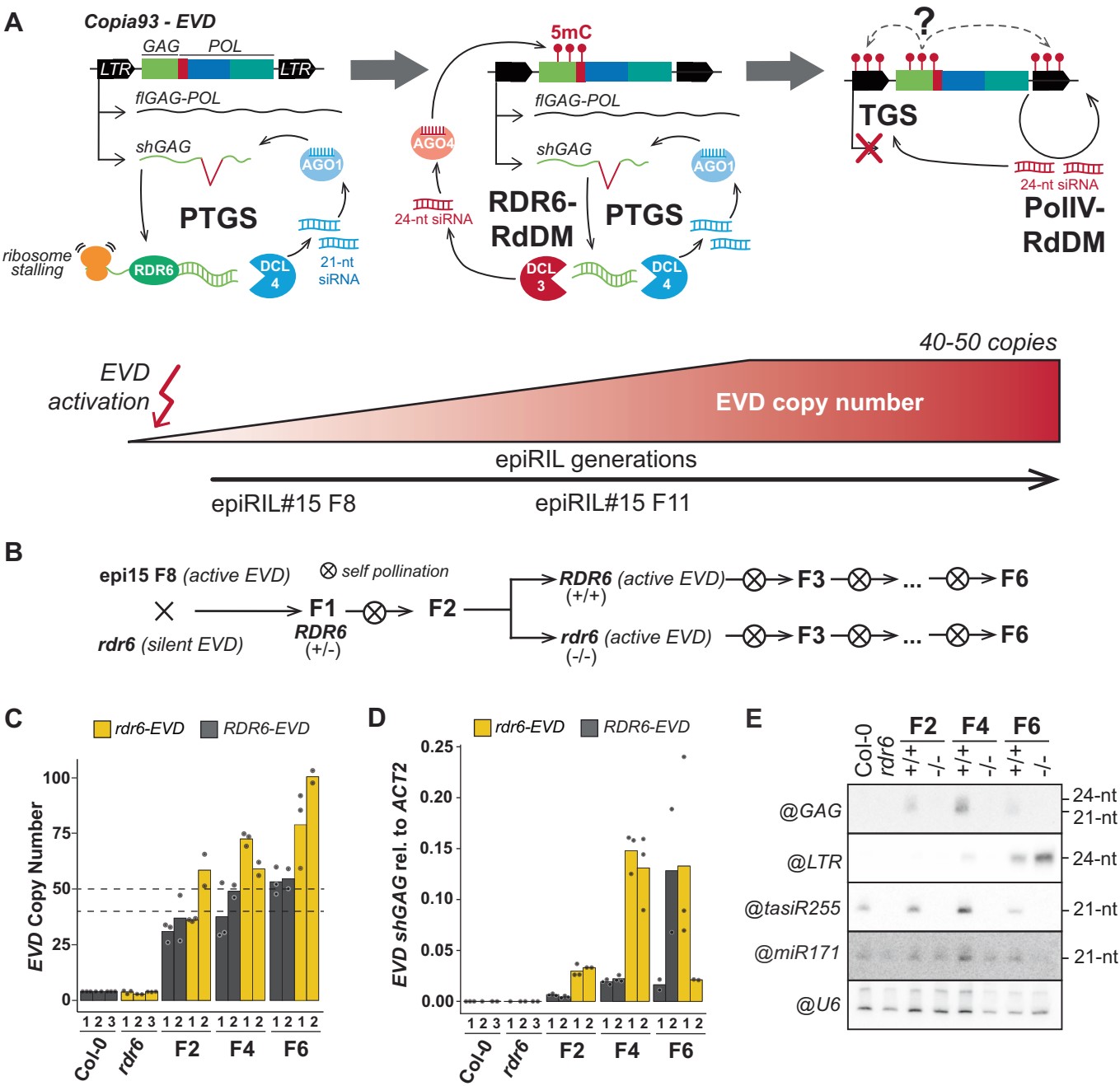

**Figure 1. Introgression and characterization of *EVD* in the *rdr6* mutant background.**

(A) Schematic representation of the three *EVD* silencing steps. Upon *EVD* reactivation, ribosome stalling during translation of *EVD shGAG* transcript triggers PTGS. 21-nt siRNA produced through RDR6 and DCL4 and loaded into AGO1. With increasing *EVD* copies across generations, the excess of dsRNA produced by RDR6 is processed by DCL3 to generate 24-nt siRNAs. Loaded into AGO4, *shGAG* siRNAs trigger DNA methylation (5mC) through RDR6-RdDM at GAG coding sequences without silencing. At 40–50 copies per genome, TGS is installed through Pol IV-RdDM, coincidentally with the appearance of DNA methylation and 24-nt siRNAs on the LTR sequences. (B) Crossing scheme to generate *rdr6* mutant lines with active *EVD*. F2 plants were genotyped to select homozygous WT and mutant RDR6 lines, propagated through selfing until the F6 generation. (C) *EVD* copy number analysis by qPCR in RDR6-*EVD* and *rdr6*-*EVD* lines at generations F2, F4, and F6 derived from two independent F1s (biological replicates), using the *EVD*-GAG sequence as target. (D) qPCR analysis of *shGAG* expression normalized to *ACT2* in *EVD*-RDR6 and *EVD*-*rdr6* lines at generations 2, 4, and 6 derived from two independent F1s (biological replicates). In (C) and (D), each biological replicate, consistent of bulks of 8–10 plants, are represented for each genotype at each generation, dots show technical replicates. (E) RNA blot analysis of *EVD* siRNAs against GAG and LTR in RDR6 and *rdr6* lines with active *EVD* at generations F2, F4, and F6. tasiR255 probe is used as control for RDR6 mutation, miR171 and snoRNA U6 are shown as loading controls. WT Col-0 and *rdr6* with no reactivated *EVD* are shown as negative control for *EVD* activity. Source data are available online for this figure.

*rdr6-EVD* (Fig. 1D). This variation was observed consistently for the expression of both *shGAG* and *flGAG-POL* isoforms (Figs. 1D and EV1), suggesting a biological rather than a technical origin. The decrease and broad variation in *EVD* expression compared to the F4 (and despite the increase in copy number) suggested that TGS had started to take place. This was expected for the *RDR6-EVD* as the bulk of F6 individuals has already exceeded the 40–50 copy number limit, previously demonstrated to lead to TGS (Marí-Ordóñez et al, 2013). However, that result was unanticipated and intriguing for the *rdr6-EVD* lines and initiated the analysis of the potential *EVD* silencing mechanism by investigating the small RNA profile by RNA blots.

Although the transition of *EVD* from PTGS to TGS has been demonstrated to take place at the individual plant level, the detection of *EVD-GAG* and *EVD-LTR* siRNA in bulked material can also be used as proxy for the assessment of *EVD* silencing stage at each generation (Marí-Ordóñez et al, 2013). As expected from the dependency of *GAG*-derived siRNA on *EVD* expression, *EVD-GAG* siRNAs were detected in *RDR6-EVD* throughout generations, mirroring *EVD* expression levels (Fig. 1E). In these plants, 24-nt *EVD-LTR* siRNAs, involved in PolIV-RdDM and transcriptional silencing of *EVD*, were first detected in the F4 and increased in the F6 generation, coinciding with the decrease of *EVD* expression and *GAG* siRNAs (Fig. 1E). In contrast, as expected in the absence of RDR6, no RDR6-derived *EVD-GAG* siRNAs and trans-acting (ta) siRNAs were detectable in *rdr6-EVD* plants, *EVD-LTR* siRNAs were present in the F6 generation (Fig. 1E). Thus, the lack of PTGS and RDR6-derived *EVD-GAG* siRNA did not prevent the appearance of *EVD-LTR* 24-nt siRNAs, previously associated with *EVD* TGS.

## RDR6-dependent *EVD-GAG* siRNAs are dispensable for *EVD* TGS

The presence of 24-nt *EVD-LTR* siRNAs, hallmark of successful PolIV-RdDM installation, indicated that silencing of *EVD* copies through TGS was likely taking place in both RDR6-*EVD* and *rdr6-EVD* backgrounds. Previous work had shown that *EVD-GAG* siRNAs (PTGS) and *EVD-LTR* 24-nt siRNAs (PolIV-RdDM) are mutually exclusive at the individual level. While both can be detected in bulks of plants, individuals with active or silenced *EVD* produce *GAG* or *LTR* siRNAs, respectively, as the transition to TGS has been shown to impact most if not all copies within one generation at the individual level (Marí-Ordóñez et al, 2013). The detection of both in RDR6-*EVD* bulks suggested that a subset of the individual plants successfully silenced *EVD* through TGS, pointing to a different silencing status in individual F6 plants within the bulked samples of both lines.

To investigate *EVD* silencing at the level of individual plants, and whether TGS had taken place in the absence of RDR6, ten *RDR6-EVD* and ten *rdr6-EVD* individuals from the F6 generations plants were selected. In *RDR6-EVD*, 24-nt *EVD-LTR* siRNAs were detected in all but two individuals, #2 and #9, where only *EVD-GAG* siRNAs were present (Fig. 2A). The presence of 24-nt *EVD-LTR* siRNAs was associated with a lower expression of *EVD* (Fig. 2B). Similarly, 24-nt *EVD-LTR* siRNAs were also detected in 6 out of 10 *rdr6-EVD* individuals (Fig. 2A) and the presence of 24-nt

siRNAs correlated with a corresponding loss of *EVD* expression (Fig. 2B). Hence, silencing of *EVD* and production of associated *LTR* 24-nt siRNAs can take place in absence of RDR6-dependent *GAG* siRNAs.

**A**

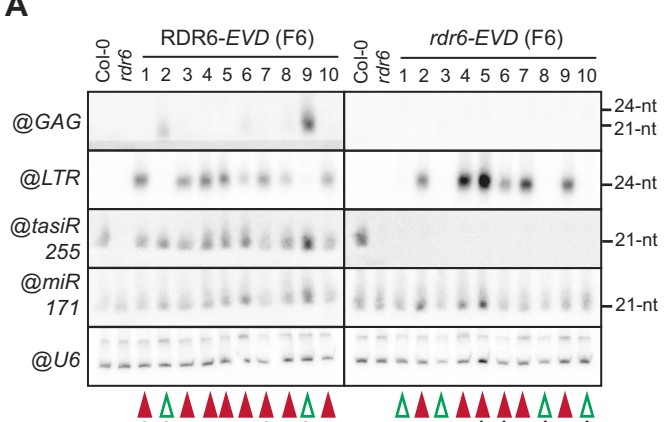

**B**

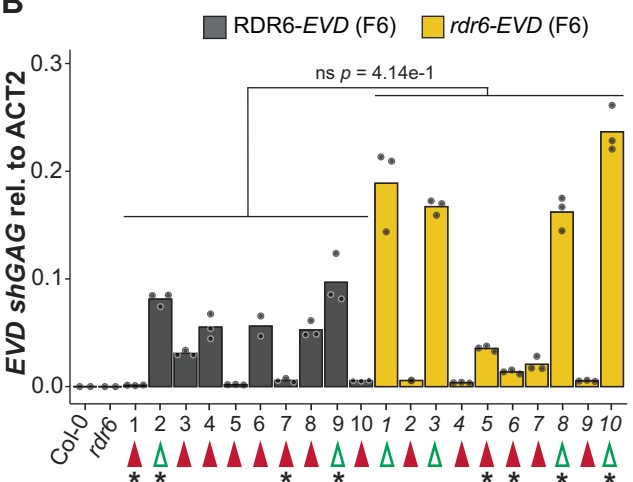

**C**

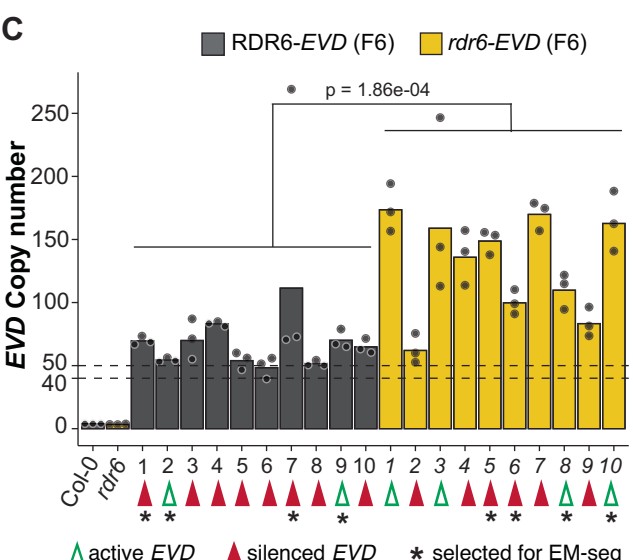

△ active *EVD*  ▲ silenced *EVD*  \* selected for EM-seq

◄ **Figure 2.  Characterization of *EVD* silencing status in *RDR6*- and *rdr6-EVD* F6 individuals.**

(A) RNA blot analysis of *EVD* siRNAs against GAG and LTR in 10 F6 individual plants of RDR6-*EVD* and *rdr6-EVD* lines. tasiR255 probe is used as control for RDR6 mutation, miR171 and snoRNA U6 are shown as loading controls. WT Col-0 and *rdr6* with no reactivated *EVD* are shown as negative control for *EVD* activity. (B) Analysis of *EVD* shGAG expression of the same individuals investigated in A by qPCR, normalized to *ACT2*. (C) *EVD* copy number analysis by qPCR of the same individuals investigated in (A) and (B), using the *EVD*-GAG sequence as target. In (B) and (C), qPCR technical replicates for each sample are represented by dots. *p*-values for two-sided t-test between indicated samples are shown. Differences are considered statistically significant if *p* < 0.05 (5.00e−2) or non-significant (ns) if *p* ≥ 0.05. Green and red arrows indicate individuals with active and silenced *EVD* copies, respectively, selected individuals for subsequent EM-sequencing are indicated by an asterisk in (A–C). Source data are available online for this figure.

## *EVD* silencing in the absence of RDR6 does not correlate with copy number

*EVD* switch to TGS has been experimentally established at around 40–50 copies in both *met1* and *ddm1* epiRILs, coinciding with the upper limit of natural variation for *COPIA93* copies found within *Arabidopsis* ecotypes (Marí-Ordóñez et al, 2013; Quadrana et al, 2016). To assess whether a similar threshold applied in the absence of RDR6, we quantified *EVD* copy number in the same *RDR6-EVD* and *rdr6-EVD* F6 individuals used for *EVD* siRNAs and expression.

In agreement with the copy number threshold above which *EVD* TGS takes place, most *RDR6-EVD* individuals displayed homogenous copy number only slightly above this range. In *rdr6-EVD* plants, however, *EVD* silencing had taken place at a more variable copy number (Fig. 2C). Many plants displayed higher copy number, as previously observed in the bulk analysis (Fig. 1C); some individuals had switched to TGS at copy numbers just above the threshold (*rdr6-EVD* #2), while others did so at a copy number well above 100 (*rdr6-EVD* #4, 5, 7) (Fig. 2C). Nonetheless, *EVD* remained active in other individuals with copy numbers above 150 (*rdr6-EVD* #1, 3, 10) (Fig. 2C). Consequently, while PTGS seemed to facilitate the establishment of TGS in a reliable manner, in the absence of RDR6 activity, no clear copy number threshold for TGS installation was observed. Thus, in the *rdr6* background, silencing of active *EVD* might be stochastic once the 40–50 copy number threshold is exceeded or depend on other factors not considered so far.

## DNA methylation of *EVD-GAG* is dispensable for the transition to TGS

The copy number threshold is likely determined by the point at which the cumulative *EVD* expression from all new insertions causes DCL3 to process *shGAG* RDR6-derived dsRNA to initiate RDR6-RdDM (Marí-Ordóñez et al, 2013). Although *EVD* switch to TGS took place in absence of RDR6 and associated *GAG* siRNAs, we could not rule out the processing of *shGAG* transcripts by one of the other *Arabidopsis* RDR proteins, producing siRNA levels undetectable by Northern blots, but sufficient to induce the *EVD-GAG* DNA methylation believed to initiate PolIV-RdDM.

To corroborate the installation of PolIV-RdDM at *EVD LTRs* and address whether DNA methylation was independently of

RDR6-generated siRNAs still deposited at *EVD-GAG*, or another coding region, *EVD* DNA methylation was assessed by whole-genome Enzymatic Methyl-sequencing (EM-seq) (Feng et al, 2020) on F6 individuals before and after the transition to TGS. We selected lines with active *EVD* (two *RDR6-EVD* and two *rdr6-EVD* individuals with high *EVD* expression and no 24-nt *EVD-LTR* siRNAs), as well as lines with silenced *EVD* (two RDR6-*EVD* and two *rdr6-EVD* individuals producing 24-nt *EVD-LTR* siRNAs and low *EVD* expression) (Fig. 2A,B; marked by asterisks). EM-seq on wild-type (*Col-0*) and *rdr6* plants was performed as control for endogenous *EVD* methylation in the respective background. Overall, a 30–50X genome coverage with conversion rates above 99.8% was obtained for all samples (Fig. EV2A–C). Because short-read sequencing technology makes it challenging to estimate methylation levels along individual *EVD* insertions (~5 kb), EM-seq reads were mapped to a single fictitious *EVD* locus to estimate average methylation levels.

In *RDR6-EVD* with active *EVD*, low levels of *GAG* methylation in all cytosine contexts were observed. In contrast, the equivalent *rdr6-EVD* lines displayed near absence of DNA methylation (Fig. 3A,B). Furthermore, DNA methylation increased upon *EVD* silencing in *RDR6-EVD* but remained low after the *EVD* silencing in *rdr6-EVD* (Fig. 3A), indicating that absence of RDR6 was sufficient to abolish *EVD-GAG* DNA methylation, not only during *EVD* proliferation, but also after silencing. In addition, DNA methylation levels in *rdr6-EVD* lines remained low across other *EVD* coding regions in *rdr6-EVD* lines with active or silenced *EVD*, compared to *RDR6-EVD* (Fig. EV2D,E), ruling out that the switch to TGS is triggered through non-canonical-RdDM activity at other regions in the absence of RDR6.

We next investigated DNA methylation at *EVD-LTRs* to confirm that presence of *LTR* 24-nt siRNA were bona fide indicators of PolIV-RdDM and the switch to *EVD* TGS. In both *RDR6-EVD* and *rdr6-EVD* lines with active *EVD*, DNA methylation at the *LTRs* was very low in all three contexts, while methylation levels were increased in those with silenced *EVD* (Fig. 3C). Furthermore, methylation levels at CHG and specifically at CHH were higher than those of the parental *EVD* copy in the wild-type Col-0 and *rdr6* controls (Figs. 3C and EV2F), confirming the successful installation of PolIV-RdDM following the *EVD* burst, in contrast to RdDM-independent DNA methylation maintenance of *EVD* prior to its reactivation. To further validate that the switch to TGS through PolIV-RdDM installation had taken place, we inspected the methylation status of the *EVD*-derived solo-*LTR* present in the promoter of *RECOGNITION OF PERONOSPORA PARASITICA 4* (*RPP4*, *AT4G16860*), referred to as *RPP4 solo_LTR* hereafter, which can be methylated in trans by *EVD-LTR* 24-nt siRNAs following an *EVD* de novo silencing event (Marí-Ordóñez et al, 2013). Indeed, *EVD solo-LTR* was only methylated in the *RDR6-EVD* and *rdr6-EVD* lines where *EVD* had been silenced (Fig. 3D).

Hence, the switch to TGS through PolIV-RdDM does neither require RDR6-RdDM nor the deposition of DNA methylation on *EVD-GAG* or other coding regions, to successfully silence *EVD*.

## EM-seq paired-end sequencing data allows mapping of new *EVD* insertions

Once PolIV-RdDM is established, *EVD* silencing and associated deposition of DNA methylation impacts new copies genome-wide

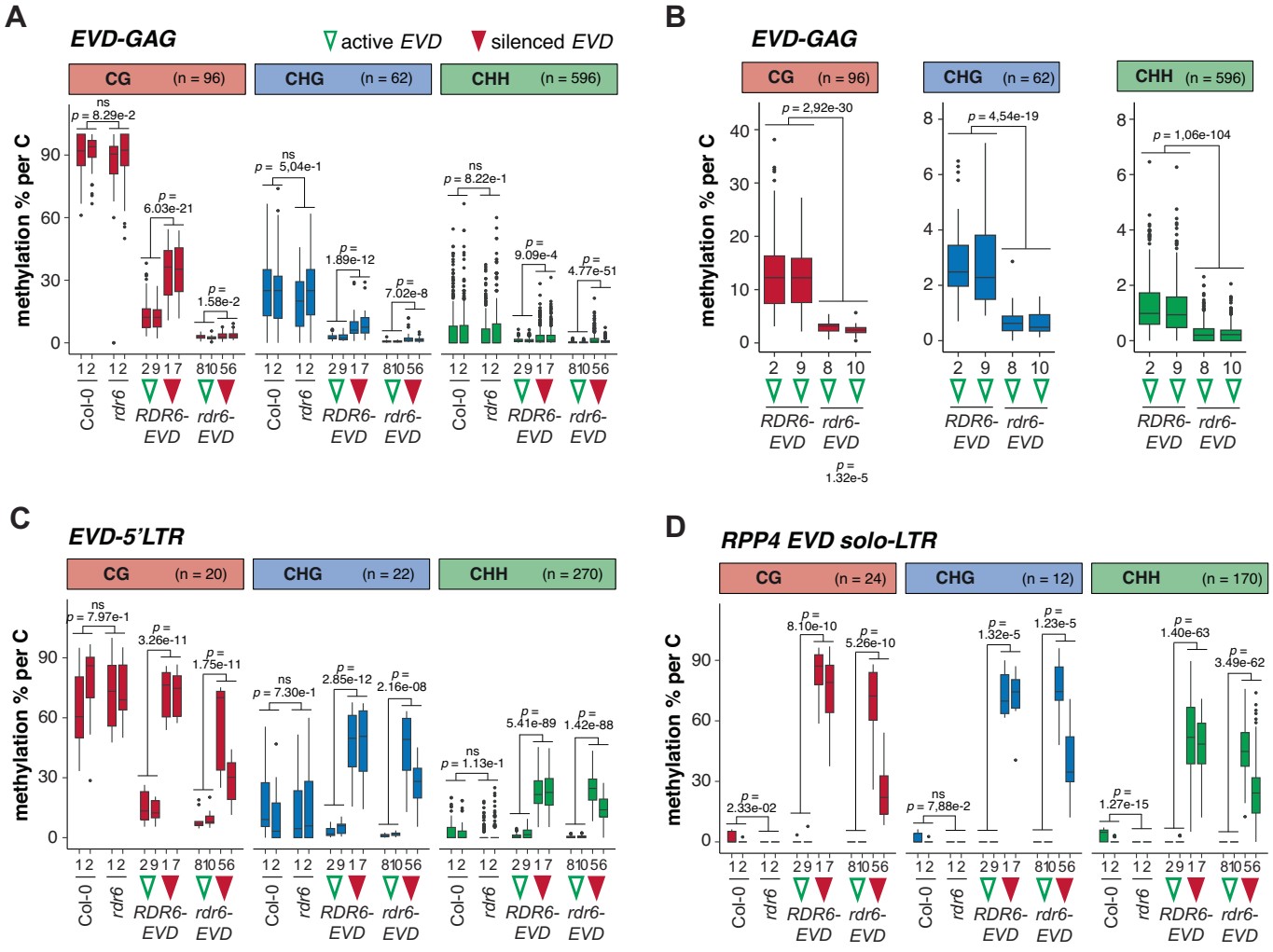

**Figure 3.  DNA methylation of active and silenced *EVD* in *RDR6*- and *rdr6-EVD* lines.**

EM-seq analysis of DNA methylation (as methylation % per cytosine) in CG, CHG, and CHH contexts in WT Col-0, *rdr6* and in *RDR6*- and *rdr6-EVD* F6 individuals with active and silenced *EVD* (marked with empty green and filled red arrowheads, respectively, numbers indicate same individuals as in Fig. 2) in: (**A**) *EVD-GAG*; (**B**) *EVD-GAG* but only in *RDR6*- and *rdr6-EVD* F6 individuals with active *EVD*; (**C**) *EVD-5'LTR*; and (**D**) *EVD solo-LTR* in *RPP4* (AT4G16869) promoter. In all panels, *n* indicates the number of cytosines analyzed for each context per sample. In all boxplots: median is indicated by a solid bar, the boxes extend from the first to the third quartile and whiskers reach to the furthest values within 1.5 times the interquartile range. Dots indicate outliers, as data points outside of the above range. Wilcoxon rank sum test adjusted *p*-values between indicated groups of samples are shown. Differences are considered statistically significant if *p* < 0.05 (5.00e−2) or non-significant (ns) if *p* ≥ 0.05. Source data are available online for this figure.

through the trans-activity of *EVD LTR* 24-nt siRNAs (Marí-Ordóñez et al, 2013). However, in *EVD*-silenced *rdr6-EVD* lines, *LTR* methylation levels were lower in the CG context for line #5 and in all contexts for line #6 than those in *RDR6-EVD* (Figs. 3C and EV2F). To investigate the homogeneity of DNA methylation at individual *EVD* insertions, and to ask whether it is influenced by the absence of RDR6-RdDM, we took advantage of discordant paired-read mates in our EM-seq data, where one of the read mates mapped to *EVD LTRs* and the other elsewhere in the genome, to identify and locate new *EVD* insertions (Gilly et al, 2014; Stuart et al, 2016; Quadrana et al, 2016; 2019) (Figs. 4A and EV3A) and to assess their methylation levels. Only new *EVD* insertions supported by three or more discordant paired-read mates from both of *EVD LTRs* were considered. Simultaneously, concordant paired-read mates mapping to *EVD* were used to independently estimate *EVD* copy numbers through the increase in

*EVD* sequencing coverage due to additional insertions in *RDR6*- and *rdr6-EVD* relative to the Col-0 reference genome (Yoon et al, 2009; Quadrana et al, 2016) (Figs. 4A and EV3B).

While no new *EVD* insertions were obtained in Col-0 and *rdr6* controls, estimation methods yielded consistent increased *EVD* copy numbers in lines where *EVD* had proliferated (Fig. 4B). As shown above (Fig. 2C), more *EVD* insertions were found in *rdr6-EVD* than in *RDR6-EVD* (Fig. 4B). In all lines, 75% or more of new insertions were mapped to chromosome arms (Figs. 4C and EV3C), as expected from the integration preference of *EVD* into gene-rich regions (Quadrana et al, 2019). Furthermore, most new *EVD* insertions caused short sequence duplications, known as target site duplications (TSD), of 5–8 bp (Fig. 4D). This falls within the TSD range expected for LTR-RTE in plants, specifically a 5-bp TSDs as obtained here for *EVD* (Quadrana et al, 2016; Orozco-Arias et al,

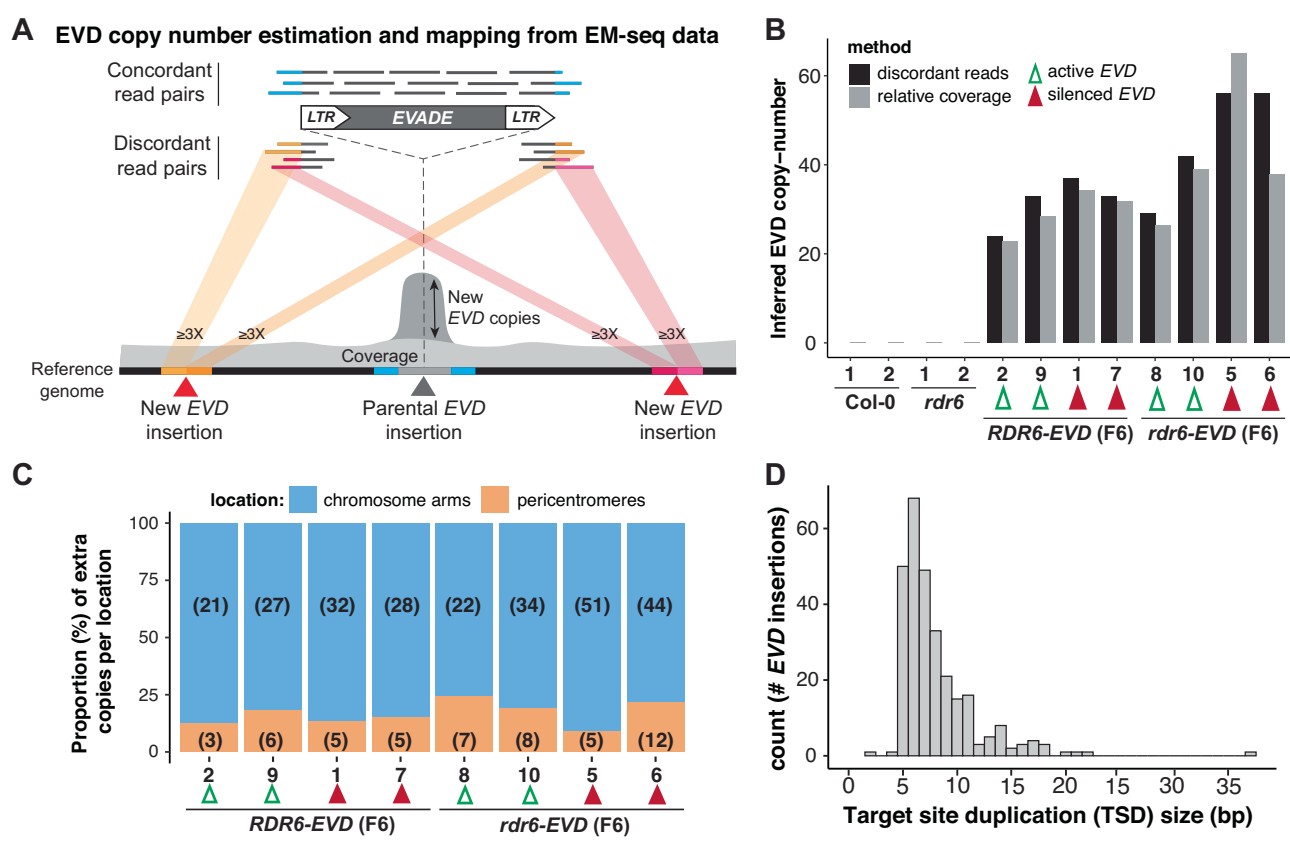

**Figure 4.   Quantification of new *EVD* insertions in *RDR6*- and *rdr6-EVD* lines from EM-seq data.**

(A) Schematic representation of the strategy used to quantify and map new *EVD* insertions from EM-seq data. *EVD* copy number was estimated using: (i) the increased *EVD* coverage of concordant paired read mates in EM-seq data, consequence of *EVD* transposition or, (ii) mapping new *EVD* insertions through discordant read pairs mapping to *EVD* and elsewhere in the genome. New insertions had to be supported by at least three discordant read pairs from each border to be considered. (B) Inferred *EVD* copy number using either relative coverage or discordant reads in WT Col-0, *rdr6* and in *RDR6*- and *rdr6-EVD* F6 individuals with active and silenced *EVD* (marked with empty green and filled red arrowheads, respectively, numbers indicate same individuals as in Figs. 2 and 3). (C) Number and relative proportion (in %) of mapped new *EVD* insertions in pericentromeric or chromosomic arm locations in each indicated sample. Numbers of new *EVD* insertions at each location are indicated in brackets. (D) Histogram of the size (in bp) distribution of target site duplications at all new *EVD* insertions. Source data are available online for this figure.

2019; Jedlicka et al, 2019; Roquis et al, 2021). Therefore, the new *EVD* insertions mapped from EM-seq data displayed features of bona fide new *EVD* transposition events.

We noticed that *EVD* copy numbers quantified from EM-seq data were lower than the quantification by qPCR. This discrepancy might originate from inaccuracy of qPCR quantifications, including potential amplification of *EVD* extra-chromosomal cDNA, or from limited discordant read-mates coverage in the EM-seq data. Nonetheless, defined insertions identified in all samples allowed to investigate methylation levels at individual *LTRs* of new *EVD* insertions.

## DNA methylation is not homogeneously deposited across new *EVD* insertions

As observed in our global analysis of *EVD* DNA methylation (Fig. 3), both *LTRs* of individual new *EVD* insertions gained DNA methylation in all contexts in *EVD*-silenced lines compared to those with active *EVD*. However, the *LTR* DNA methylation levels of individual *EVD* insertions were less homogeneous than expected, displaying a broad range in all contexts. Notwithstanding, we

observed that the levels of DNA methylation in the three contexts were more heterogenous in *rdr6-EVD* lines than in the equivalent *RDR6-EVD* ones (Fig. 5A). This was most remarkable in the CG context, where most insertions in *RDR6-EVD* lines with silenced *EVD* displayed CG methylation levels above 50%, whereas several insertions in *rdr6-EVD* display lower or no methylation, especially in the *rdr6-EVD* line #6. A similar trend was observed in the CHG and CHH contexts, where some insertions displayed lower or no methylation in the *rdr6-EVD* lines after the switch to PolIV-RdDM (Fig. 5A).

To investigate if the observed variation was the result of different methylation levels of individual insertions or the two *LTRs* from the same insertion being independently methylated, the correlation between 5′ and 3′ *LTR* DNA methylation for each insertion was assessed. While in the *RDR6-EVD* lines methylation between the two *LTRs* displayed a weak but positive correlation in the three contexts, in absence of RDR6 such correlation was lower for all three contexts (Fig. 5B). Again, this effect was more pronounced for CG methylation. Although we observed variation in *RDR6-EVD* silenced lines between 5′ and 3′ *LTR* methylation for a given insertion, in most cases DNA methylation remained above

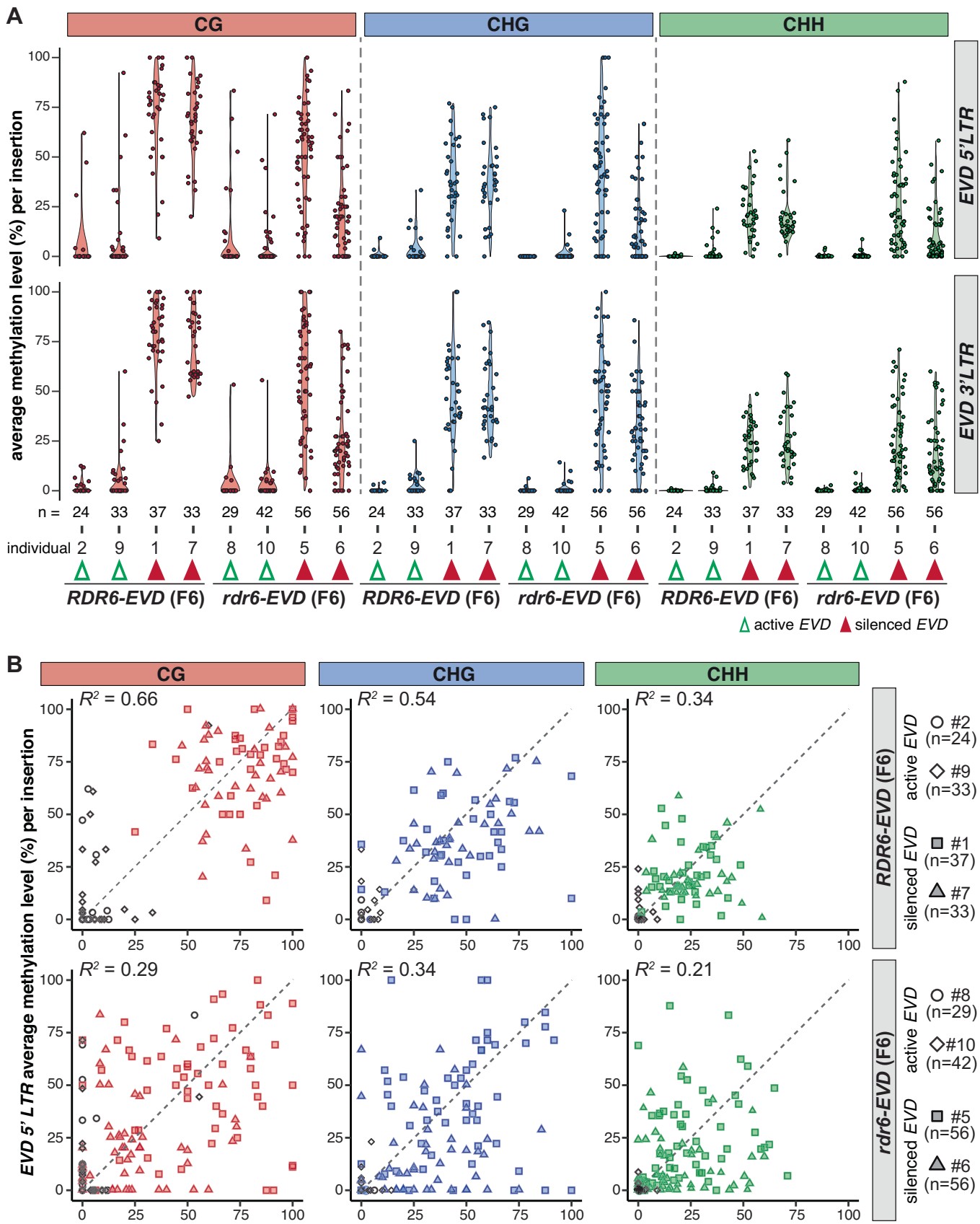

**Figure 5.** *LTR* DNA methylation of individual new *EVD* insertions in *RDR6-* and *rdr6-EVD* lines.

(A) 5′ and 3′*LTR* average DNA methylation levels in each cytosine context for individual new *EVD* insertions in *RDR6-* and *rdr6-EVD* F6 individuals with active and silenced *EVD*. (B) Correlation between 5′ and 3′*LTR* average DNA methylation levels in each cytosine context for individual *EVD* insertions in *RDR6-* and *rdr6-EVD* F6 individuals. $R^2$ indicates correlation coefficient between 5′ and 3′ *LTR* methylation levels within individual *EVD* insertions. Dashed line shows $R^2 = 1$. Empty black symbols indicate *EVD* copies from individuals with active *EVD* and color-filled symbols from individuals with silenced *EVD*. In all panels, number of *EVD* insertions analyzed per individual correspond to those indicated in Fig. 4C. In all panels, n indicates the number of *EVD* insertions analyzed per sample. Source data are available online for this figure.

50% in both *LTR*s. However, in *rdr6-EVD* lines, differences in CG methylation between *LTR*s of the same insertion were more pronounced, with several *EVD* insertions displaying high CG levels in one *LTR* but not the other. No bias for preferent methylation of either 5′ or 3′ *LTR* was observed (Fig. 5B).

Thus, in absence of RDR6 DNA methylation levels were not only less homogenous between de novo silenced *EVD* insertions (Fig. 5A), but also between *LTR*s of the same insertion (Fig. 5B). This could be a consequence of the absence of potential priming for the switch to TGS provided by *EVD-GAG* DNA methylation but might also reflect a difference in the timing of the switch. *EVD-LTR* 24-nt siRNAs were already detected at the 4th generation in *RDR6-EVD* but not in *rdr6-EVD* (Fig. 1E). Thus, in the later, *EVD* likely had been under PolIV-RdDM for less generations. This might be the case especially in the *rdr6-EVD* #6 line, where *RPP4 solo_LTR* methylation levels, which depends on *EVD-LTR*-derived 24-nt siRNAs, are also lower than in the other *EVD*-silenced lines (Fig. 3D). Nonetheless, the gain of DNA methylation at most new *EVD* insertions further supports that PolIV-RdDM gets installed at *EVD-LTR*s in the absence of RDR6-RdDM.

## RdR6-RdDM is insufficient for *EVD* silencing in the absence of PolIV-RdDM

The above results indicated that, independently of RDR6-RdDM and *EVD-GAG* methylation, *EVD* silencing likely took place through the initiation of PolIV-RdDM at *EVD-LTR*s. While both PolIV and PolV are essential for PolIV-RdDM, only PolV is required for RDR6-RdDM, if siRNAs are provided through PTGS (Nuthikattu et al, 2013; McCue et al, 2015; Taochy et al, 2019; Sigman et al, 2021). Hence, to test the dependency of *EVD* TGS in PolIV-RdDM and, at the same time, explore if RDR6-RdDM could lead to TGS independently of PolIV-RdDM, we introduced active *EVD* in the mutants *nrpd1* (PolIV largest subunit mutant) and *nrpe1* (PolV largest subunit mutant), following the same strategy as for *rdr6*. This time, however, to ensure the presence of *EVD-GAG* methylation before the loss of RdDM, an epi15 F11 generation plant, already undergoing RDR6-RdDM (Marí-Ordóñez et al, 2013) (Fig. 1A), was used as *EVD* donor. Again, wild-type and mutant plants were selected in the F2 and propagated to F6 (Fig. EV4A).

While lines carrying wild-type or mutant alleles were indistinguishable with respect to *EVD* copy number or expression in the F2 (Fig. 6A,B), *EVD* reached higher copy numbers, surpassing the 40–50 copy number threshold, in the following generations of both mutants (*nrpd1-EVD* and *nrpe1-EVD*) than in the lines carrying the corresponding wild-type alleles (*NRPD1-EVD* and *NRPE1-EVD*) (Fig. 6A). Similarly to the results obtained with *RDR6-EVD* lines, once the threshold was reached in lines with wild-type alleles, *EVD* expression was reduced. However, in both mutant backgrounds, *EVD* remained transcriptionally active (Fig. 6B), indicating that TGS was not installed in the absence of PolIV-RdDM.

Furthermore, investigation of *EVD* small RNA patterns in three independent bulks of plants at the F6 generation confirmed the presence of *EVD-LTR* 24-nt siRNAs in both *NRPD1-* and *NRPE1-EVD* lines. On the contrary, in *nrpd1-* and *nrpe1-EVD* lines, only *EVD-GAG* siRNAs were detected (Fig. 6C,D), consistent with *EVD* expression triggering PTGS.

Methylation of *EVD* was assessed in F6 bulks by Sanger sequencing of PCR amplicons from bisulfite-treated DNA (BS-PCR). Although this PCR-based method does not discern between integrated and extra-chromosomal DNA (ecDNA) integration intermediates in lines with active *EVD*, it allowed us to estimate overall *EVD* methylation levels. Regarding *EVD-GAG*, CG methylation was high in both wild type and mutant lines, as expected according to MET1 maintenance and previous exposure to RDR6-RdDM (Fig. EV4B–E). CHG and CHH methylation was present in both WT and mutant lines, albeit higher in *NRPD1-* and *NRPE1-EVD* lines than in their mutant equivalents (Fig. EV4B–E), probably due to the increase in methylation previously observed after the installation of TGS (Fig. 3A). Both *nrpd1-* and *nrpe1-EVD* lines displayed similar CHG methylation levels likely inherited from the epi15 F11 and maintained independently of siRNAs. However, CHH methylation was higher in *nrpd1-EVD* than in *nrpe1-EVD* lines (Fig. EV4B–E), in agreement with the absence of RDR6-RdDM in *nrpe1* but not in *nrpd1* mutants. Therefore, RdD6-RdDM seems to still be depositing CHH methylation in the *nrpd1-EVD* lines.

We next investigated the methylation status of *EVD-LTR*. As anticipated from the different epigenetic regulation of *EVD* before and after its mobilization and silencing, the presence of 24-nt siRNAs in *NRPD-* and *NRPE-EVD* lines correlated with increased DNA methylation levels, where CHG and CHH methylation levels surpassed those found in WT and mutant controls (Figs. 6E,F and EV4F,G). Furthermore, no loss of *EVD* DNA methylation, relative to Col-0, was observed neither in *nrpd1* nor *nrpe1* control samples, confirming the absence of PolIV-RdDM regulation at the parental *EVD* insertion (Fig. 6E,F). In contrast, in *nrpd1-* and *nrpe1-EVD* lines, DNA methylation in all contexts remained low in conformity with the absence of *EVD-LTR* 24-nt siRNAs and TGS (Fig. 6E,F). However, we noticed that DNA methylation in all contexts was higher in *nrpd1-EVD* than in *nrpe1-EVD*, suggesting that continuous RDR6-RdDM activity might cause weak DNA methylation in *EVD-LTR*. Still, as the BS-PCR used here amplifies both integrated and ecDNA *EVD* copies, a similar EM-seq strategy as used for the *EVD-rdr6* F6 individuals will be required in the future to obtain *LTR* methylation levels of individual insertions in *nrpd1-* and *nrpe1-EVD* lines. Nonetheless, in the absence of PolIV-RdDM, *EVD* remained transcriptionally active up to the F6 generation despite the continuous action of PTGS and, in the case of *nrpd1-EVD*, RDR6-RdDM. Hence, PolIV-RdDM is required for de novo TGS initiation independently of RDR6-RdDM.

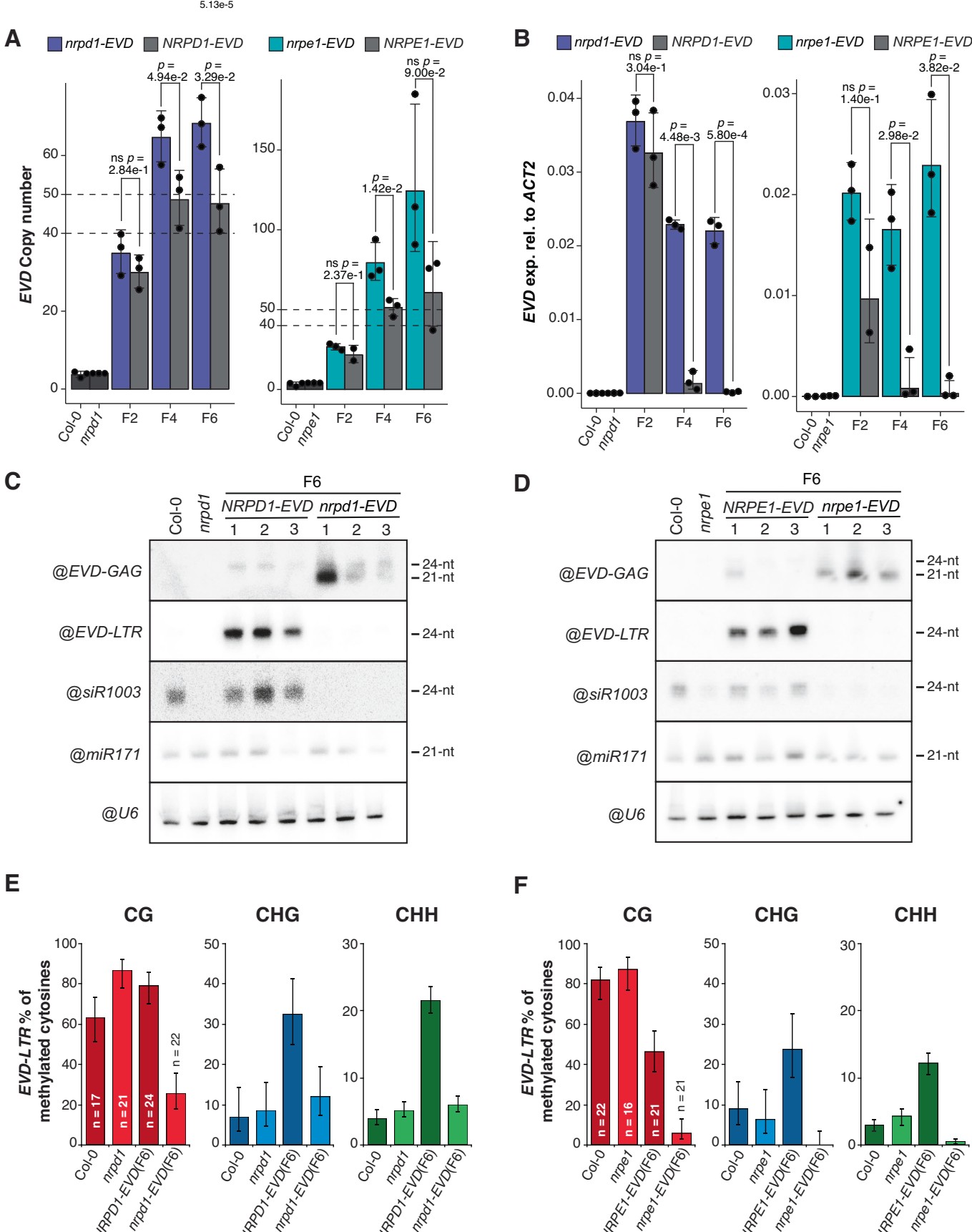

Figure 6. Characterization of *EVD* proliferation and silencing in RdDM mutants.

(A) *EVD* copy number analysis by qPCR in *NRPD1-EVD* and *NRPE1-EVD* lines, in both WT (gray) and mutant (colored) backgrounds, at generations F2, F4, and F6 derived from three independent F1s (biological replicates), using the *EVD*-GAG sequence as target. (B) qPCR analysis of *shGAG* expression normalized to *ACT2* in *NRPD1-EVD* and *NRPE1-EVD* lines, in both WT (gray) and mutant (colored) backgrounds, at generations F2, F4, and F6 derived from three independent F1s (biological replicates). In (A) and (B), biological replicates (bulks of 8–10 plants each) are individually represented by dots. Error bars show standard error of the mean in (A) and (B). *p*-values for two-sided t-test between indicated samples are shown. Differences are considered statistically significant if *p* < 0.05 (5.00e−2) or non-significant (ns) if *p* ≥ 0.05. (C, D) RNA blot analysis of *EVD* siRNAs against GAG and LTR in the F6 generation of 3 independent WT and mutant lines of *NRPD1-EVD* (C) and *NRPE1-EVD* (D). WT Col-0 and *nrpd1* with no reactivated *EVD* are shown as negative control for *EVD* activity. siR1003 probe is used as control for *NRPD1* and *NRPE1* mutations, miR171 and snoRNA U6 are shown as loading controls. (E, F) % methylated cytosines (C) by bisulfite-PCR DNA methylation analysis at *EVD*-LTR sequences in the F6 generation of *NRPD1-EVD* (E) and *NRPE1-EVD* (F) lines, in both WT (darker shade) and mutant (lighter shade) backgrounds. Col-0, *nrp1d* and *nrpe1* were used as controls. *n*: number of clones analyzed. Error bars represent 95% confidence Wilson score intervals of the % of methylated cytosines (C) in each context (CG, CHG, CHH). Source data are available online for this figure.

## *EVD* antisense nested insertions and transposition into RdDM loci are potential TGS initiation events in absence of RdR6-RdDM

Given that PolIV-RdDM requires pre-existing epigenetic marks for its recruitment, *EVD-LTR* 24-nt siRNAs might be a consequence of a preceding *EVD* silencing event rather than the cause. To gain further insights into the trigger of *EVD* TGS in the absence of RDR6-RdDM, we investigated presumed triggers of TGS in the *rdr6-EVD* lines.

First, we attempted to explore the potential presence of *LTR* hairpins as the initial source of *LTR* siRNAs. TE-derived hairpins can arise as the result of genomic rearrangements between TE sequences or tandem/nested insertions and drive the silencing of homologous TEs (Slotkin et al, 2005). To do so, we initially examined the EM-seq data in search for discordant reads where the mate pairs will both map to *EVD-LTR* but in opposite orientations (Fig. EV5). We did not succeed to identify any read pair mate in such configuration (Fig. EV5). However, detecting *LTR* hairpins through such approach is technically limited by the insert size of the EM-seq library (~300–700 bp) as any pair of *EVD-LTR*s in antisense orientation further away than that will not be captured by such strategy with the EM-seq data. Therefore, as we could not detect the presence of two LTRs in close proximity, we expanded the search to paired read mates mapping to *LTR* and to antisense *EVD* sequences to identify potential nested antisense insertion configurations (Fig. 7A). Although such read pair mates were found in several individuals (Fig. 7A), we set a threshold of at least 2 read mates at each *LTR* to confidentially identify *EVD* nested insertions within itself and discard sequencing artifacts. Using this strategy, only one antisense nested insertion of *EVD* into itself was found in the *rdr6-EVD #5* individual, in which *EVD* had been silenced, but not in any other sample (Fig. 7A,B). In the resulting *EVD* locus, the *LTR* of the initial insertion are in antisense to the *LTR* of the incoming *EVD* insertion. With the *LTR*s being 2562 and 1942 bp apart, this configuration can potentially result in *LTR* hairpin formation by transcription from the initial insertion into the antisense *3′LTR* from the nested one or all the way through the antisense insertion to generate a second hairpin (Fig. 7B). Alternatively, convergent transcription driven by the *5′LTR* of each of the insertions might result in the formation of dsRNA (Fig. 7B). Both cases can potentially trigger *LTR* siRNAs to initiate *EVD* TGS (Fig. 7B) and will require further investigation. However, such *EVD* antisense nested insertion was only confidentially found in one of the two silenced *rdr6-EVD* lines.

As an alternative trigger, we next investigated the local DNA methylation status of new *EVD* insertion sites. Transposition into a pre-existing RdDM locus might suffice to initiate PolIV-RdDM on

an *EVD* insertion and spread to other copies through the action of 24-nt siRNAs. Therefore, we aimed to determine the methylation status of *EVD* landing sites prior to transposition in order to investigate if integration within methylated loci was a common feature in individuals with silenced *EVD*. However, extensive DNA methylation variation has been reported for the *met1*-epiRILs (Reinders et al, 2009). As the epiRIL#15 was used as the parental line carrying active *EVD* to generate the *RDR6-* and *rdr6-EVD* lines (Fig. 1B), their methylomes might differ from that of the wild-type. Hence, we used the EM-seq data to assess the methylation of individual landing sites for new *EVD* insertions by examining their DNA methylation in individuals without *EVD* at the corresponding locations (Fig. 7C). Fixed *EVD* insertions, present in all individuals sequenced, were discarded as no information about the methylation status of the region prior to insertion could be obtained. Only polymorphic *EVD* insertions were considered for the analysis (Fig. 7C). In addition, as absence of RDR6 has been shown to impact DNA methylation (Stroud et al, 2013), the methylation levels at regions without new *EVD* insertions were only assessed from individuals of the same genotype. Therefore, the methylation at positions with *EVD* insertions shared by all individuals of the same genotype were considered non-informative (Fig. 7C). Furthermore, their presence in all active and silenced *EVD* individuals within a given genotype likely disqualified them as the triggers of TGS. Regions were classified as unmethylated (5mC < 10%), mCG only (mCG > 10%, mCHG and mCHH < 10%) or non-CG methylated (mCHG, mCHH > 10%). In all samples, most new *EVD* insertions happened at unmethylated locations. *EVD* transposition events at CG and non-CG methylated loci were rare but also found in individuals with active or silenced *EVD* (Fig. 7D), suggesting that such events were not determinant of *EVD* silencing status. However, as CHH methylation can be maintained by PolIV-RdDM or independently of small RNAs through CMT2 activity, only insertions in PolIV-RdDM-dependent loci might led to 24-nt siRNAs production. Although the dependency of mCHH on PolIV-RdDM and CMT2 on a given locus might not be conserved in the *RDR6-* and *rdr6-EVD* lines given their epiRIL origin, we further subclassify non-CG methylated positions into RdDM, CMT2 or RdDM/CMT2-independent. Pathway assignment was based on the annotation performed by Sasaki and colleagues (Sasaki et al, 2022), who used a large methylome dataset from *Arabidopsis* mutants (Stroud et al, 2013) to determine locus-specific DNA methylation dependencies on different silencing pathways. No clear correlation between insertion of *EVD* in a putative PolIV-RdDM locus and its silencing was observed, as *EVD* insertions in RdDM loci were observed in an *EVD* active *RDR6-EVD* individual and only in one *EVD* silenced *rdr6-EVD* plant (#6) (Fig. 7D).

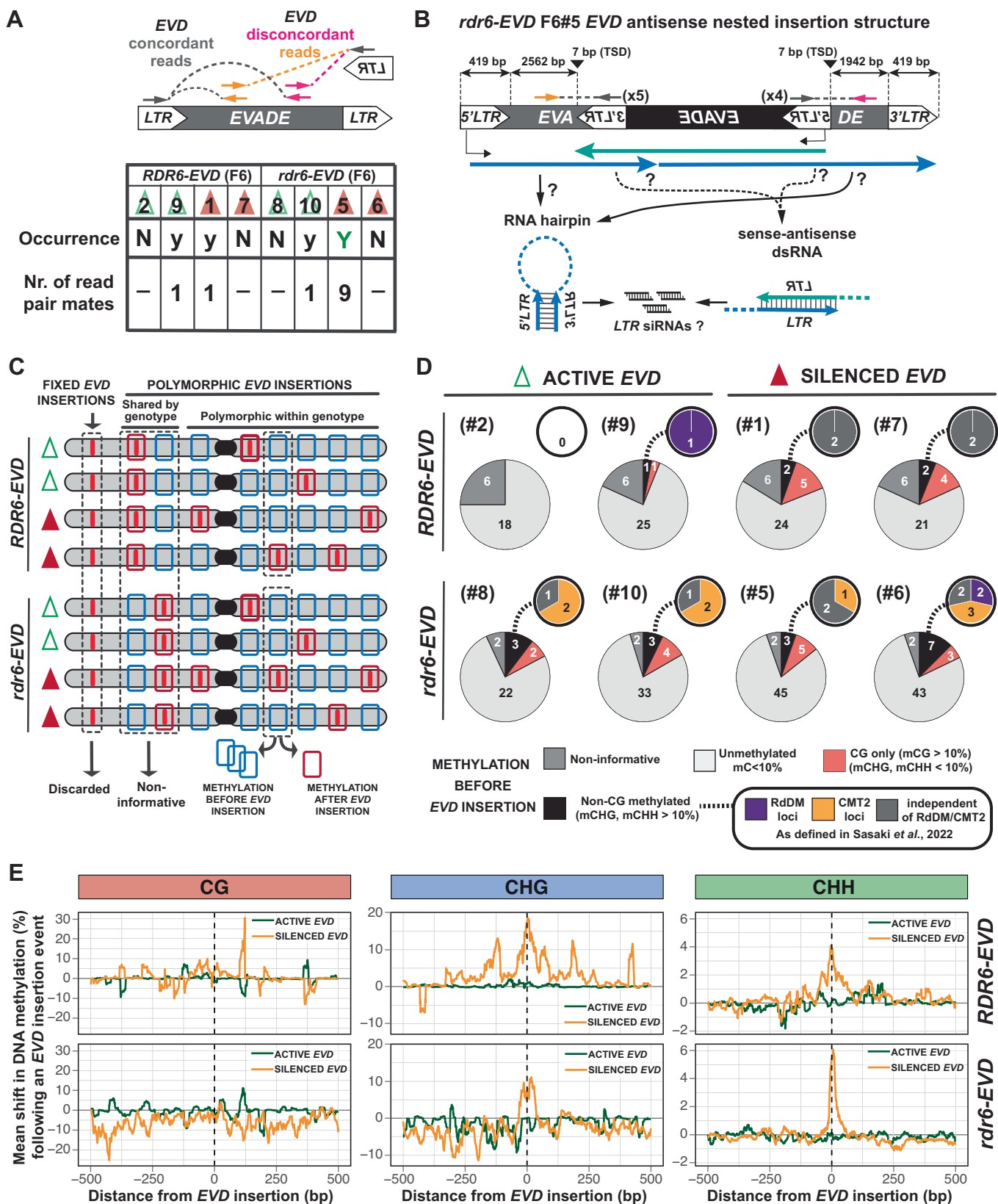

© The Author(s)

**Figure 7. Genetic and epigenetic characterization of *EVD* insertion sites.**

(A) Schematic representation of the strategy used to identify *EVD* antisense nested insertions within itself from EM-seq discordant paired read mates mapping to *EVD-LTR* and elsewhere on *EVD* coding sequence in antisense to the *LTR*. The table indicates the occurrence and number of read mates found in each sample. Arbitrary threshold of at least two paired read mates by *LTR* was set to confidentially identify an *EVD* nested insertion. N: non-occurrence, y: presence of at least one discordant paired read mate, Y: presence of discordant paired read mates and above threshold for confident presence of nested insertion. (B) Scheme of *EVD* antisense nested insertion structure based on *EVD* discordant reads in the sample *rdr6-EVD* F6 #5. Distance between *LTRs* as well as the length of the target-site duplication (TSD) caused by the insertion are indicated in base-pairs (bp). Colored arrows represent potential *EVD* transcripts within the locus. Putative sources of small RNAs are depicted below. (C) Schematic representation of methodology used to infer the methylation status of *EVD* landing sites prior to transposition using the EM-seq data. *EVD* insertions are represented by red lines. Red boxes indicate regions with new and polymorphic *EVD* insertions. Blue boxes represent the same region without the *EVD* insertion. (D) Pie charts of the distribution of insertions within non-methylated, CG-only methylated and non-CG methylated regions within each sample. Inlets represent the classification of non-CG methylated regions into RdDM-, CMT2-dependent or independent of both. Number of insertions in each category is indicated within the chart. (E) Metaplot of the mean shift in DNA methylation following a *EVD* insertion by genotype and methylation context depending on *EVD* silencing status. Source data are available online for this figure.

Incidentally, such individual did not carry the *EVD* antisense nested insertion (Fig. 7A) and remains possible that those insertions triggered the switch to TGS in the *rdr6-EVD* plant #6.

Nonetheless, these analyses are not conclusive as the sample size is too small. Further small RNA, epigenetic characterization and genome sequencing of lines with active and silenced *EVD* will be required to find whether insertions within specific epigenetic landscape or *EVD* insertion configurations are responsible for initiating the epigenetic silencing of all new *EVD* insertions in absence of RDR6-RdDM.

## *EVD* transposition and silencing has limited impact on local DNA methylation

Lastly, despite *EVD* preferential insertion within non-methylated regions (Fig. 7E), changes in local chromatin in response to transposition, and eventually leading to TGS, might had occurred. To additionally investigate if *EVD* transposition led to alterations in local DNA methylation, and whether those were influenced by the epigenetic status of *EVD*, we compared methylation levels at *EVD* insertion sites and their surroundings with and without *EVD* and before/after switch to TGS. Globally, no gains of DNA methylation in any context were observed around the insertion sites upon *EVD* transposition in plants with active *EVD*. However, methylation in CHG and CHH, but not in CG, increased at those positions following *EVD* transition to TGS (Fig. 7E). Although we noticed that CHG and CHH gains reached further away in *RDR6* than in *rdr6* lines, these did not spread far from the insertion site, likely reflecting the role of RdDM not only maintaining DNA methylation but also at preventing it from spreading outside of silenced loci boundaries (Zemach et al, 2013; Li et al, 2015).

## Discussion

### PTGS is required for *GAG* methylation but dispensable for the TGS of *EVD*

Post transcriptional gene silencing mediated by RDR6-dependent siRNAs has been hypothesized to be the initiating step of TE epigenetic silencing, in particular *EVD* (Teixeira et al, 2009; Marí-Ordóñez et al, 2013; Nuthikattu et al, 2013; McCue et al, 2015; Panda et al, 2016; Sigman et al, 2021). In this study, by investigating the silencing fate of active *EVD* in wild type and mutant *RDR6* backgrounds, we show that, although DNA methylation deposited in the *EVD-GAG* sequence is a consequence of PTGS through RDR6-RdDM and might contribute to its final silencing, in absence

of RDR6 and the associated *EVD-GAG* siRNAs, transcriptional gene silencing is still achieved through the installation of PolIV-RdDM without any prior DNA methylation at *EVD* coding sequences. Therefore, PTGS and gene body DNA methylation are dispensable for de novo *EVD* silencing.

Our results add further evidence supporting that PTGS can direct the deposition of DNA methylation (Wu et al, 2012; Nuthikattu et al, 2013; McCue et al, 2015; Taochy et al, 2019). However, RDR6-RdDM is neither necessary nor sufficient to for TGS initiation, which requires PolIV-RdDM at regulatory sequences. Previous work has shown that an immobile *EVD* overexpression transgenic system under the control of the CaMV 35S promoter (*35S:EVD*), despite triggering the same PTGS response as the endogenous *EVD*, does not transition to TGS in WT plants (Marí-Ordóñez et al, 2013; Oberlin et al, 2022). This is in line with the observation that in endogenous loci and transgenes triggering strong PTGS and undergoing RDR6-RdDM, DNA methylation is unable to spread from transcribed regions to regulatory sequences to set TGS (Taochy et al, 2019).

### PTGS might facilitate the installation of *EVD* TGS

Nonetheless, given the consistent installation of silencing in presence of PTGS, limiting *EVD* proliferation beyond 40–50 copies, and the loss of *LTR* DNA methylation homogeneity between as well as within newly silenced *EVD* insertions, RDR6-RdDM likely sensitizes *EVD* loci to facilitate TGS initiation, explaining the uniformity of *EVD* copy number at which RdDM is installed at *EVD*, compared to the stochasticity observed in *rdr6* mutants. Introduction of either active *EVD* or *35S:EVD* in *dcl2 dcl4* double mutants, results in the switch to TGS at 20–30 *EVD* copies and silencing of *35S:EVD* coupled with the production of 24-nt siRNAs from their promoters (Marí-Ordóñez et al, 2013). Hence, in the case of *EVD*, enhancing RDR6-RdDM by promoting DCL3 activity upon RDR6 dsRNA products in the absence of DCL2 and DCL4, expedites *EVD* switch from PTGS to TGS. Here, despite the lack of PolIV-RdDM, a low level of DNA methylation at *EVD-LTR* was found in RDR6-RdDM competent *nrpd1-EVD* plants. Given the close proximity of *GAG* to 5′ regulatory sequences, strong or prolonged *GAG* RDR6-RdDM might result in enough methylation adjacent to the promoter to recruit PolIV-RdDM as previously suggested (Marí-Ordóñez et al, 2013; Sigman et al, 2021). This situation may be favored for endogenous *EVD*. Although both endogenous *EVD* and *35S:EVD* produce high RDR6-dependent siRNA levels, in contrast to the ubiquitously expressed *35S* promoter, *EVD-LTR* drives expression in only a few cells (Marí-

Ordóñez et al, 2013), where intracellular siRNAs levels might be high enough to promote DNA methylation spreading into the nearby *LTR*. However, as the BS-PCR used here to measure *EVD* DNA methylation in *polIV* and *polV* mutant backgrounds does not provide information about the methylation status of individual insertions and it captures both integrated and ecDNA *EVD* copies, a similar EM-seq strategy as used for the *EVD-rdr6* F6 individuals will have to be applied in the future. This will allow to obtain a better resolution of the *LTR* methylation levels of individual insertions in *nrpd1-* and *nrpe1-EVD* lines to gain further insights in the role of RDR6-RdDM in priming or sensitizing *EVD* insertions for the switch to TGS.

## PTGS potentially acts as an *EVD* copy number control system

Alternatively, although not mutually exclusive with the priming role of RDR6-RdDM in the switch to TGS, we propose that PTGS triggered during translation can act as a mechanism to regulate *EVD* proliferation until the switch to TGS takes place. Mutations in *RDR6* lead to increased *EVD shGAG* mRNA, protein and VLP levels, resulting in increased transposition (this study and (Lee et al, 2021; Oberlin et al, 2022)). Although the mechanisms triggering *shGAG* ribosome stalling and cleavage has not yet been elucidated, given that PTGS is not commonly triggered by most TEs undergoing translation in *Arabidopsis* (Oberlin et al, 2022), it is possible that *EVD* hijacks the PTGS pathway to mediate copy number control (CNC) and regulate its own transposition rate. A variety of CNC factors have been found to be self-encoded by TEs, such as peptides to inhibit VLP formation or antisense ORFs to hamper reverse transcription, as means to minimize genomic damage on their host caused by over proliferation (Matsuda and Garfinkel, 2009; Cottee et al, 2021). However, *EVD* also reached high copy numbers in *nrpd1-* and *nrpe1-EVD* lines, despite the continuous action of PTGS. A full understanding of the impact of PTGS in *EVD* transposition and host recognition and defense mechanisms will require further investigation of *EVD* activity in genetic backgrounds defective for both PTGS and RdDM.

## PolIV-RdDM is essential for *EVD* control and de novo silencing

Recruitment of RdDM to *EVD-LTRs* independently of RDR6-RdDM, together with the absence of *EVD* TGS in either *nrpd1* or *nrpe1* backgrounds, indicates that PolIV-RdDM is essential for de novo *EVD* silencing during its genome colonization. Although absence of RdDM has little impact in the reactivation of silenced TEs in the *Arabidopsis* genome (He et al, 2021; 2022), a role for RdDM in controlling TE propagation has been previously shown for the heat-responsive LTR-retrotransposon (LTR-RTE) *ONSEN*, for which proliferation and copy number following induction is increased in *NRPD1* mutants (Ito et al, 2011; Matsunaga et al, 2012; Hayashi et al, 2020; Niu et al, 2022). Furthermore, genetic variation in the RdDM components *RDR2* and *NRPE1* has been associated with variation in TE content within *Arabidopsis* natural populations (Baduel et al, 2021; Sasaki et al, 2022; Jiang et al, 2023). Therefore, PolIV-RdDM might play a major role in de novo TE silencing besides its function as a DNA methylation maintenance pathway.

Coincidentally, RdDM at long, young, and potentially functional TEs, operates at their edges (Zemach et al, 2013; Stroud et al, 2013; 2014), similarly to the patterns found at *EVD* after the transition to TGS independently of PTGS. As mentioned previously, most LTR-RTEs intact enough to be translation-competent do not trigger PTGS when transcriptionally active (Oberlin et al, 2022). Hence, the phenomena of RdDM installation at flanking *LTRs* in absence of PTGS observed here for *EVD* might represent a more general mechanism of de novo silencing of LTR-RTEs.

## Potential mechanisms of TGS initiation in absence of PTGS

In absence of PTGS and the initial deposition of DNA methylation within the *EVD-GAG* sequence, the mechanism(s) triggering or initiating epigenetic silencing on *EVD* remain obscure as we have not identified a common cause. Apart from TE activity, several mechanisms have been shown to recruit RdDM or trigger the deposition of DNA methylation. Some of which have been explored in this work and will be further discussed here as they could explain the stochasticity of silencing observed.

For example, double-stranded DNA breaks (DSBs) have been shown to trigger the production of siRNAs and promote the deposition of DNA methylation at the borders of the break points in *Arabidopsis* (Wei et al, 2012; Schalk et al, 2016; 2017; Du et al, 2022). In addition, little is known about the chromatin landscape of newly integrated TE copies, which might contain histone variants or histone modifications predisposing for DSB-induced silencing (Yelagandula et al, 2014; Lorković et al, 2017; Osakabe et al, 2018; Bourguet et al, 2021). However, no changes in DNA methylation were observed around new insertion sites in individuals with active *EVD*. Moreover, such mechanism would imply that new insertions become individually silenced as they integrate, which is incompatible with the low levels of DNA methylation in lines with active *EVD* and its coordinated increase once the switch to TGS takes place. A significant number of simultaneous transposition events generating excessive DNA damage might be required to elicit a silencing response. Nonetheless, *EVD* does remain active in some individuals despite a high copy number and expression levels.

Given the arbitrary copy number at which TGS took place in *rdr6-EVD* lines, silencing might therefore depend on more stochastic events. Due to trans-activity of *EVD-LTR* 24-nt siRNAs, the initiation of RdDM in one or few *EVD* copies might suffice to spread silencing across all new active insertions. RNA hairpins resulting of TE tandem or nested insertions and acting as source of 24-nt siRNAs has been shown to mediate the epigenetic silencing of the *Mutator* TE in maize (Slotkin et al, 2003). Though this mechanism was previously deemed unlikely to trigger *EVD* TGS in *RDR6* wild-type backgrounds (Marí-Ordóñez et al, 2013), increased transposition upon absence of PTGS might enhance the chances of hairpin formation. We have indeed detected event of an antisense *EVD* nested insertion into itself, potentially leading to the formation of *LTR* hairpins, in a *rdr6-EVD* individual where *EVD* has switched to TGS. However, we did not test if such *EVD* locus did become a source of siRNAs to initiate *EVD* switch to TGS. Nonetheless, RNA hairpins are generally processed into several siRNA sizes (Zilberman et al, 2004; Fusaro et al, 2006), and *LTR* siRNAs in such individual were predominantly 24-nt long.

Alternatively, and analogously to TE silencing triggered by transposition within piRNA clusters in *Drosophila melanogaster* (Brennecke et al, 2007; Goriaux et al, 2014; Guida et al, 2016), increased transposition might increase the probability of integration events within pre-existing heterochromatin domains, subjecting one or few *EVD* copies to RdDM. However, no increase in absolute or relative pericentromeric insertions was observed in *rdr6-EVD* lines that had switch to *EVD* TGS and, although *EVD* insertions within regions with non-CG methylation were observed, those were not exclusive to individuals that had switch to TGS. Transposition within putative RdDM loci were observed in one *rdr6-EVD* individual with silenced *EVD* but as well in one *RDR6-EVD* individual with active *EVD*. Therefore, although transposition events leading to the formation of *EVD* loci with the potential to trigger small RNAs independently of RDR6 or within RdDM loci can occur, a more detailed study with increased population size, better characterization of the chromatin environment and small RNAs in *EVD* landing sites will be required in the future to properly explore if transposition within certain chromatin environments or loci already subjected to RdDM can initiate *EVD* TGS.

In addition, the repetitiveness of *EVD* during a transposition burst, might led to changes in the three-dimensional chromatin organization forming, or bringing *EVD* to, chromatin interaction clusters associated with silencing (Grob et al, 2014; Grob and Grossniklaus, 2019). Furthermore, such repetitiveness might cause genome instability through non-allelic homologous recombination events (Sammarco et al, 2022) that trigger the silencing of *EVD*. Future investigation of the presence of genome rearrangements and chromatin conformation changes before and after *EVD* switch to TGS should help elucidate whether such changes are taking place and are indeed linked to de novo *EVD* silencing.

Several of the above mechanisms might be favored by the presence of LTRs. Although single LTRs from exogenous TEs have been shown to trigger RdDM when transformed into *Arabidopsis* (Fultz and Slotkin, 2017), they are found as duplicated sequences at both ends of intact LTR-RTEs. Such arrangement might help the formation of LTR hairpins through tandem (either in head-to-head or tail-to-tail configurations) or nested *EVD* insertions, as observed for multicopy T-DNA events (Neve et al, 1997; Buck et al, 1999). Moreover, *EVD-LTRs* being twice the number of *EVD* insertions might increase non-allelic recombination rates between *LTRs*. Furthermore, in *Arabidopsis*, it has been shown that the structure of a locus can influence its epigenetic fate using a transgenic selection marker flanked by tandemly duplicated promoter sequences, mimicking an LTR-RTE configuration. Such transgenic locus can be found in either active or silenced state (epialleles). The silenced epiallele, characterized by the be presence of 24-nt siRNA from the repeated regions (like *EVD*), has been shown to induce silencing of the active one in an RdDM-dependent manner when both are crossed with each other (Scheid et al, 2003; Foerster et al, 2011; Bente et al, 2021). Deletion of the duplicated region at the end of the transgene from the silent epiallele lead to a decrease in its trans-silencing ability, while its removal from the active epiallele strongly impaired its capacity to become silenced (Bente et al, 2021). Hence, this type of sequence configuration might contribute to spread and stabilize TGS across *EVD* insertions once one of them becomes targeted by RdDM.

Finally, RdDM has been implicated in antiviral defense against Geminiviruses, plant single-stranded circular DNA viruses replicating in the nucleus (Al-Kaff et al, 1998; Raja et al, 2008). Linear and circular extra chromosomal DNA (ecDNA and eccDNA,

respectively), products of reverse transcription, have been detected upon expression off several LTR-RTEs, including *EVD* (Lanciano et al, 2017; Mann et al, 2022; Niu et al, 2022). In absence of PTGS, *EVD* ecDNA and eccDNA content has been shown to increase in plant tissues (Lee et al, 2020; Zhang et al, 2023). Although it remains to be determined whether RdDM can act on TE ecDNA or eccDNA, they harbor the potential to become RdDM targets for the initiation of TGS on integrated *EVD* copies.

In conclusion, our study reveals that, while PTGS activity contributes to limit TE proliferation ensuring the faithful setting of TGS, PTGS alone is insufficient to explain the de novo initiation of epigenetic silencing of an active proliferative TE in the *Arabidopsis* genome. TGS initiation in absence of PTGS is more stochastic and, although we have not identified a common trigger event, some of the potential causes identified here will increase the chance of occurring as more *EVD* copies accumulate in the genome. Besides those, several of the above-mentioned phenomena, including those not investigated here, alone or in combination, could contribute to de novo silencing of *EVD* and TEs, and deserve future investigation to gain insights into the mechanisms of defense against genomic parasites.

# Methods

## Plant material and growth conditions

Plants were grown on soil at 21 °C, in 16 h light/8 h dark cycle with an LED light intensity of 85 µM m$^{-2}$ s$^{-1}$. After germination, seedlings were transplanted at a rate of 1 plant per pot. Plants were genotyped by PCR as soon as a tip of a young rosette leaf could be harvested (see Reagents and Tools table for a list of primers used). Inflorescences (closed flower buds, no visible petals) were harvested, flash frozen in liquid nitrogen and stored at −70 °C. Cauline leaves used for *EVD* copy number quantification were harvested on ice and immediately stored at −70 °C.

## Estimation of *EVD* expression and copy number by qPCR

Total RNA was extracted from 6 to 8 closed inflorescences ground to a fine powder by TRIzol™ according to manufacturer's instructions and precipitated in 1 volume of cold isopropanol. Total RNA was treated with RNase-free DNase I and cDNA synthesis performed using the RevertAid Reverse Transcription Kit with random hexamer primers. qPCR reactions were performed in a total volume of 10 µl using the KAPA SYBR FAST qPCR Master Mix (2X) with low ROX reference dye according to the manufacturer's instruction and run on a QuantStudio 5 Real-Time PCR System. Each reaction was performed in 2–3 technical replicates. Relative expression was calculated as fold change of the ratio of target of interest and ACT2 (AT3G18780). For *EVD* copy number estimations, DNA was extracted from frozen ground cauline leaves using Edwards buffer (200 mM Tris pH 8, 200 mM NaCl, 25 mM EDTA, 0.5% (v/v) SDS). Samples were precipitated in 1 volume of 70% ethanol and diluted at 1/100 in double-distilled water. *EVD* copy number was estimated by absolute quantification from the ΔΔCt EVD and ACT2 levels, normalized by their inherent copy numbers of two in WT plants, respectively. Oligonucleotides used are listed in the Tools and Reagents table. Data analysis was done in Excel (Microsoft) and R. Plots were generated using

**Reagents and tools table**

| Reagent/Resource | Reference or Source | Identifier or Catalog Number |
|---|---|---|
| **Experimental Models** | | |
| *A. thaliana* Col-0 (WT) | (Alonso et al, 2003) | NASC ID: N60000; ABRC ID: CS60000 |
| *A. thaliana rdr6-15* | (Fahlgren et al, 2006) | NASC ID: N879578; ABRC ID: CS879578 (SAIL_617_H07) |
| *A. thaliana* epi15 F8 | (Reinders et al, 2009) | N/A |
| *A. thaliana* epi15 F8 x *rdr6-15* | (Oberlin et al, 2022) | N/A |
| *A. thaliana* epi15 F11 | (Marí-Ordóñez et al, 2013) | N/A |
| *A. thaliana nrpd1a-4 (nrpd1)* | (Pontier et al, 2005) | NASC ID: N583051; ABRC ID: SALK_083051 |
| *A. thaliana nrpd1b-11 (nrpe1)* | (Pontier et al, 2005) | NASC ID: N529919; ABRC ID: SALK_029919 |
| *A. thaliana* epi15 F11 x *nrpd1* | This work | N/A |
| *A. thaliana* epi15 F11 x *nrpe1* | This work | N/A |
| **Oligonucleotides and other sequence-based reagents** | | |
| **PCR/qPCR** | | |

| Use | Target | Sequence 5′ → 3′ | Notes |
|---|---|---|---|
| Copy number/ Expression levels | ACT2 F | GCACCCTGTTCTTCTTACCG | |
| Copy number/ Expression levels | ACT2 R | AACCCTCGTAGATTGGCACA | |
| Copy number | EVD-GAG F | TTTGACCCGCGTGTTTGAAG | |
| Copy number | EVD-GAG R | AATCTTCGGGTCAAGCGTTC | |
| Copy number | EVD-IN F | CCGGAGAACAAAGAAGCAAGC | |
| Copy number | EVD-IN R | AATGTGCGGTTCTTGGTTGG | |
| Expression levels | shGAG F | GTTGGTTGCTACATCCACACCT | |
| Expression levels | shGAG R | TTTTCCCGTCTCAATATCCGGATT | |
| BiS-PCR | EVD-LTR F | GGATATGTATTATAAGAGAGAGTGGGTYGAATATATG | |
| BiS-PCR | EVD-LTR R | TTTATAARCATAAAAACATAATCTTATRCTCTAATACCATA | |
| BiS-PCR | EVD-3′GAG F | GTGTTTGAAGTGGAAGAAGGYGATTAAYGAATTAA | |
| BiS-PCR | EVD-3′GAG R | ATAACCCRACTTAACTTTRCTCCTCATAAATTTCTTAAA | |
| Genotyping | SAIL_LB3 | tagcatctgaatttcataaccaatctcgatacac | T-DNA BP primer for genotyping *rdr6-15* |
| Genotyping | rdr6-15 LP | TGAATCCATTCCTGAACAAGC | *mut rdr6-15*: BP x LP |
| Genotyping | rdr6-15 RP | CAATGCAACCTCATCTTGGATG | WT *RDR6*: LP x RP |
| Genotyping | SALK_LBa1 | TGGTTCACGTAGTGGGCCATCG | T-DNA BP for genotyping *nrpd1a-4* and *nrpd1b-11* |
| Genotyping | NRPD1B-11 LP | CAAAGTGGTGATGCATGGAGG | *mut nrpd1b-11*: BP x LP |
| Genotyping | NRPD1B-11 RP | ATGTAAATTTTGGAAGTCGGC | WT *NRPD1A-11*: LP x RP |
| Genotyping | NRPD1A-4 LP | GCACGGGTTCGAATACGGG | *mut nrpd1a-4*: BP x LP |
| Genotyping | NRPD1A-4 RP | GTATCTGACACCGCGGACTC | WT *NRPD1A-4*: LP x RP |
| **Probes** | | |

| Use | Target | Sequence 5′ → 3′ | Notes |
|---|---|---|---|
| Northern blots | EVD-LTR F | ATGATGCTCGAGAGTGCGACAAGATCGATGTAGGT | PCR probe |
| Northern blots | EVD-LTR R | TACAATTCCGCATATTCTTTCATGGTATCAGAGCATA | PCR probe |
| Northern blots | EVD-GAG F | TAAGTCAAGAAGACTTAGAGTTTA | PCR probe |
| Northern blots | EVD-GAG R | AAGAAACTCATGAGGAGCAAAGT | PCR probe |

| Reagent/Resource | | Reference or Source | Identifier or Catalog Number |
|---|---|---|---|
| Northern blots | siR1003 | ATGCCAAGTTTGGCCTCACGGTCT | Oligo probe |
| Northern blots | tasi255 | TACGCTATGTTGGACTTAGAA | Oligo probe |
| Northern blots | miR171 | GATATTGGCGCGGCTCAATCA | Oligo probe |
| Northern blots | U6 | AGGGGCCATGCTAATCTTCTC | Oligo probe |
| **Chemicals, Enzymes and other reagents** | | | |
| TRIzol™ | | Thermo Fisher Scientific | #15596018 |
| DNase I | | Thermo Fisher Scientific | #EN0521 |
| RevertAid Reverse Transcription Kit | | Thermo Fisher Scientific | #K1622 |
| KAPA SYBR FAST qPCR Master Mix (2X) | | Sigma-Aldrich/Merck | #KK4602 |
| 2X Novex™ TBE-Urea sample buffer | | Thermo Fisher Scientific | #LLC6876 |
| Proto+Urea protein and sequencing gel sytem | | National Diagnostics | EC-830 + EC-835 + EC-840 |
| Hybond-NX Nylon membrane | | Cytivia | #RPN303T |
| 1-methylimidazol | | Merk | #M50834 |
| EDC (N-(3-dimethylaminopropyl)-N'-ethylcarbodiimide hydrochloride) | | Merk | #E7750 |
| $[\alpha\text{-}^{32}P]$-dCTP | | Hartmann Analytic | #SRP-305 |
| $[\gamma\text{-}^{32}P]$-ATP | | Hartmann Analytic | #SRP-501 |
| Prime-it II Random Primer Labelling Kit | | Agilent | #300385 |
| T4 Polynucleotide kinase | | Thermo Fisher Scientific | #EK0031 |
| illustra Microspin G-50 columns | | Cytivia | #27533001 |
| PerfectHyb | | Sigma-Aldrich/Merck | #H7033 |
| DNeasy plant kit | | Qiagen | #69204 |
| NEBNext® Enzymatic Methyl-seq Kit | | NEB | #M7634 |
| EZ DNA Methylation-Gold Kit | | Zymo Research | #D5005 |
| CloneJET PCR Cloning Kit | | Thermo Fisher Scientific | #K1231 |
| **Software** | | | |
| The code developed for this publication can be found here: | | https://github.com/GregViallefond/Trasser2024/ | |
| Excel (16.86) | | Microsoft | N/A |
| R (4.4.1) | | R Core Team; https://www.R-project.org | N/A |
| ggplot2 (3.5.1) | | (Wickham, 2016) https://cloud.r-project.org/web/packages/ggplot2/index.html | N/A |
| TrimGalore (0.6.2) | | https://github.com/FelixKrueger/TrimGalore | N/A |
| Bismark (v0.24.2) | | (Krueger and Andrews, 2011) https://github.com/FelixKrueger/Bismark | N/A |
| SAMtools (1.20) | | (Li et al, 2009) https://github.com/samtools/samtools | N/A |
| bedtools (2.28) | | https://github.com/arq5x/bedtools2 | N/A |
| picard (3.2.0) | | https://broadinstitute.github.io/picard/ | N/A |
| Kismeth | | (Gruntman et al, 2008) https://katahdin.girihlet.com/kismeth/revpage.pl | N/A |
| **Other** | | | |
| QuantStudio 5 Real-Time PCR System | | Thermo Fisher Scientific | A28575 |
| Typhoon FLA 9500 | | GE Healthcare | N/A |
| NovaSeq SP | | Illumina | N/A |

ggplot2. Statistical significance was calculated using pairwise Student's t-test. No blinding was performed.

## Small RNA blot analysis

Total RNA was extracted from 6 to 8 closed inflorescences ground to a fine powder by TRIzol™ according to manufacturer's instructions and precipitated in 1 volume of cold isopropanol. 10–20 µg of total RNA were mixed with an equal volume of 2X Novex™ TBE-Urea sample buffer and resolved on a 17.5% National Diagnostics polyacrylamide-ureal gel according to manufacturer's instructions, followed by electroblotting on a Hybond-NX Nylon membrane and chemical crosslinking (12.5 M 1-methylimidazol, 31.25 mg/mL EDC (N-(3-dimethylaminopropyl)-N'-ethylcarbodiimide hydrochloride)) according to (Pall and Hamilton, 2008). PCR probes were labeled with [α-³²P]-dCTP using the Prime-it II Random Primer Labelling Kit and purified on illustra Microspin G-50 columns according to manufacturer's instructions. Oligo probes were labeled using [γ-³²P]-ATP, using T4 Polynucleotide kinase according to manufacturer's instructions and purified on illustra Microspin G25 columns. Hybridization was performed in PerfectHyb hybridization buffer for PCR probes, or in Church buffer (7% SDS, 0.5 M $Na_2HPO_4/NaH_2PO_4$ pH 7.2, 1 mM EDTA) for oligonucleotide probes. Following overnight hybridization at 42 °C in a rotary oven, membranes were washed three times at 50 °C with 2X SCC, 0.1% (v/v) SDS. Detection was performed with a Typhoon FLA 9500. Oligonucleotide used for probes are listen in the Tools and Reagents table.

## Methylation analysis by EM-Seq

DNA was extracted from closed inflorescences from single plants, using the DNeasy plant kit according to manufacturer's instructions. Libraries were prepared using the NEBNext® Enzymatic Methyl-seq Kit following the manufacturer's instructions. Sequencing was performed on an Illumina NovaSeq SP to produce paired-end reads of 150 bp. Sequenced reads were quality filtered and adapter trimmed using TrimGalore version 0.6.2. Enzymatic-converted reads were aligned to the TAIR10 Arabiopsis genome using Bismark version v0.24.2 with bismark (--unmapped --ambiguous --maxins 700) PCR/optical duplicates were removed using *deduplicate_bismark* and the resulting BAM files were further processed to generate cytosine files *bismark_methylation_extractor* (--paired-end --cytosine_report --CX_context --comprehensive --no_overlap). Weighted average methylation (Schultz et al, 2012) was calculated and for cytosines laying on each *EVD* domain. Conversion rates were assessed by calculating the average methylation level for reads mapping to the unmethylated chloroplast (Cokus et al, 2008). Statistical testing was done in R using the function wilcox_test from the rstatix package. P-values were adjusted for multiple testing using Benjamini & Hochberg. Meaning of significance: ****$p < 10^{-4}$, ***$p < 10^{-3}$, **$p < 0.01$, *$p < 0.05$, ns $p \geq 0.05$. No blinding was performed.

## EVD copy number estimation and mapping of new insertions from EM-seq

### Coverage-based copy number estimation
Depth of coverage at every base between the 5′UTR and 3′UTR of reference EVD insertion (Chr5:5630399-5634902), as well as on two ~10 kb intervals upstream and downstream of EVD (Chr5:5619977-5629277, Chr5:5636011-5645311) was calculated for each paired-end BAM using *samtools depth* (-aa). The ratio between mean depth of coverage on EVD and mean depth of coverage on the upstream and downstream intervals was calculated for each sample. The mean inverse ratio from Col-0 and *rdr6-15* samples only carrying the reference insertion were used as a normalization factor in every other sample.

### Discordant read mate pairs-based mapping of EVD insertions
For the identification of new *EVD* insertions, an existing protocol for the detection of TE insertions from short-read sequencing data, was adapted for EM-seq datasets (Baduel et al, 2021). Each pair of fastq files containing unmapped mates was remapped in single-end (-s) mode using *bismark* (--local for both mates, --pbat for second in pair) to the TAIR10 genome, as well as to the EVD DNA sequence extracted from TAIR10 (Chr5:5629976-5635312) with parameters --local -D 20 -R 3 -N 1 for both mates, --pbat for second in pair. Since multi-mapping reads are automatically removed by *bismark*, part of the 3′ LTR from the extracted EVD sequence was masked using *bedtools maskfasta* (Chr5:5635050:5635156) to prevent reads from mapping exactly to both LTRs, leading to some missmapped reads but maximizing coverage. Fragments for which one mate mapped onto the extracted *EVD* sequence and the other to any other position in the genome were considered as discordant and candidate reads for identification of de novo insertions. Cases where a single read would map to both *EVD* and another genomic location were discarded.

Candidate reads were extracted using *bedtools bamtobed* and loaded into R for processing. A simple scheme was used to identify potential insertions: reads that mapped outside of *EVD* were sorted by chromosome and start coordinates, and then put into clusters based on their proximity. The directionality (relative to the reference genome) of their component reads was deduced from their strandness and their pairing (first or second in pair). The directionality of candidate reads mapping around a bona fide non-reference insertion is known as they should all point towards the insertion site. Therefore, valid clusters (comprised of $m + n$ reads) were considered those for which the first $m$ reads (upstream of the insertion site) pointed toward the 3′ end of the reference chromosome and the remaining $n$ reads (downstream of the insertion site) pointed towards the 5′ end of the reference chromosome ($m \geq 3$, $n \geq 3$), from those the insertion site was inferred as located between the end coordinate of the $m$th read and the start coordinates of the $m+1$st read. Similarly, paired-end mates of the first $m$ reads (or last $n$) should all share the same directionality and additionally, the first $m$ and the last $n$ reads display opposite directionality. Any cluster that did not adhere to these rules was discarded. Within each cluster, the directionality of reads mapping onto *EVD* was used to assess the strandness of the insertion, and from it each read's LTR-of-origin inferred.

For each remaining cluster, presence and size of target-site-duplication (TSD) was established by inspecting the cigar strings of the $m$th and $m+1$st reads. If the $m$th read was soft-clipped on its 5′ end and the $m+1$st read was soft-clipped on its 3′ end, and both reads overlapped each other (which was always true), then a TSD was deemed present and the size of the overlap taken as the size of the TSD. 91% of clusters showed such a pattern.

Identification of nested/tandem insertions was done similarly. In short pairs of reads mapping discordantly onto the EVD sequence, with at least one mate overlapping the 5′/3′ LTR were retrieved. For

each sample, such reads and their mapping coordinates were then visually inspected.

## Assessment of DNA methylation levels in new *EVD* insertions

For each identified new insertion, the associated cluster of reads was retrieved and LTR-of-origin determined individually for every read. In each sample, the appropriate BAM file was subsampled so it would only include the appropriate clustered reads containing LTR-of-origin information using *picard FilterSamReads*. For each sample, cluster and LTR-of-origin, a cytosine report was obtained from the subsampled BAM as specified before and a single methylation level for each cytosine context was obtained using weighted average methylation as described above was calculated and for cytosines laying on the each individual LTR.

## Bisulfite-PCR sequencing based DNA methylation analysis

DNA was extracted from 6 to 8 closed inflorescences bulked from 4 plants ground to a fine powder, using the DNeasy plant kit according to manufacturer's instructions. Bisulfite treatment was performed with the EZ DNA Methylation-Gold Kit using the manufacturer's standard protocol. DNA fragments of interest were amplified by PCR. PCR reactions and primer design was conducted according to published recommendations (Tools and Reagents table, (Henderson et al, 2010)) and cloned using CloneJET PCR Cloning Kit. Single colonies were sequenced by Sanger sequencing. Sequences were analyzed using the browser-based Kismeth software. Wilson score intervals were used to find 95% confidence intervals for the percentages of methylated sites in each context. Results shown were obtained from two independent experiments.

## Data availability

NGS data generated for this study are accessible at NCBI SRA under ID number PRJNA1111825. Accompanying code for this publication can be found here: https://github.com/GregViallefond/Trasser2024/.

The source data of this paper are collected in the following database record: biostudies:S-SCDT-10_1038-S44319-024-00304-5.

## Peer review information

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

## Acknowledgements

This work was supported by the Gregor Mendel Institute of the Austrian Academy of Sciences core funding attributed to AM-O and MN. We thank current and former members of the Marí-Ordóñez group as well as colleagues from the Gregor Mendel Institute and the Vienna BioCenter campus for useful discussions, ideas and feedback on the project and manuscript. We are grateful to the Vienna BioCenter Core Facilities (VBCF): NGS facility for EM-seq library preparation and sequencing, and the Plant Science Facility for assistance with plant work. We also thank the IMP Molecular Biology Service for providing reagents and Sanger sequencing. Lastly, we would like to acknowledge the VBC in-house COVID-19 testing facility for their efforts in providing a safe working environment during the pandemic.

## Author contributions

**Marieke Trasser**: Conceptualization; Data curation; Formal analysis; Validation; Investigation; Visualization; Methodology; Writing—original draft; Writing—review and editing. **Grégoire Bohl-Viallefond**: Formal analysis; Investigation; Visualization; Methodology. **Verónica Barragán-Borrero**: Investigation; Methodology. **Laura Diezma-Navas**: Investigation. **Lukas Loncsek**: Formal analysis; Investigation. **Magnus Nordborg**: Supervision; Writing—review and editing. **Arturo Marí-Ordóñez**: Conceptualization; Resources; Data curation; Formal analysis; Supervision; Funding acquisition; Validation; Visualization; Writing—original draft; Project administration; Writing—review and editing.

Source data underlying figure panels in this paper may have individual authorship assigned. Where available, figure panel/source data authorship is listed in the following database record: biostudies:S-SCDT-10_1038-S44319-024-00304-5.

## Disclosure and competing interests statement

The authors declare no competing interests.

# Expanded View Figures

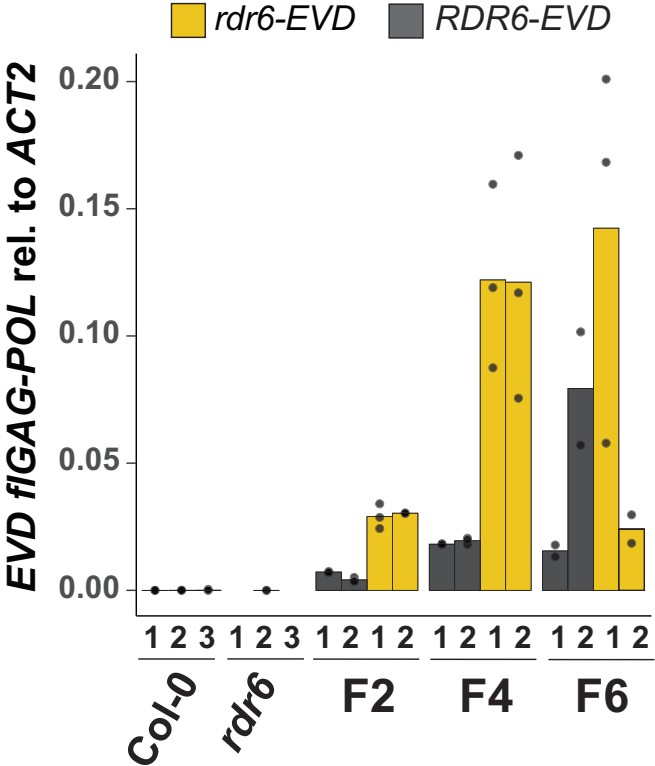

**Figure EV1.** *EVD GAG-POL* **expression in RDR6 wild-type and mutant backgrounds.**

(**A**) qPCR analysis of *flGAG-POL* expression normalized to *ACT2* in *EVD*-RDR6 and *EVD-rdr6* lines at generations 2, 4, and 6 derived from two independent F1s (biological replicates). Each biological replicate, consistent of bulks of 8–10 plants, are represented for each genotype at each generation, dots show technical replicates. Source data are available online for this figure.

# A

| sample | conversion rate |
|---|---|
| Col-0 #1 | 99,857 % |
| Col-0 #2 | 99,866 % |
| *rdr6-15* #1 | 99,847 % |
| *rdr6-15* #2 | 99,861 % |
| *RDR6-EVD* #2 | 99,850 % |
| *RDR6-EVD* #9 | 99,870 % |
| *RDR6-EVD* #1 | 99,878 % |
| *RDR6-EVD* #7 | 99,862 % |
| *rdr6-EVD* #8 | 99,864 % |
| *rdr6-EVD* #10 | 99,830 % |
| *rdr6-EVD* #5 | 99,880 % |
| *rdr6-EVD* #6 | 99,865 % |

# B

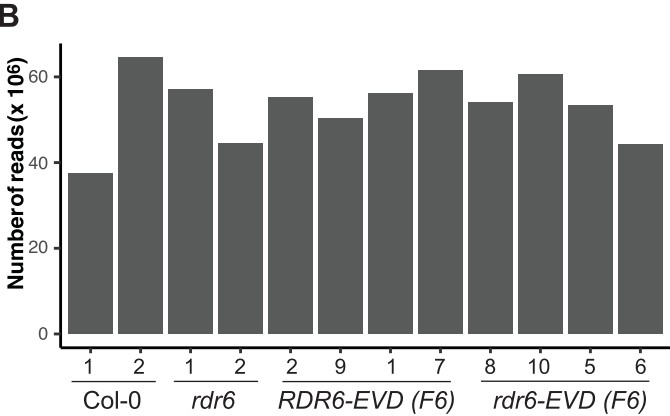

# C

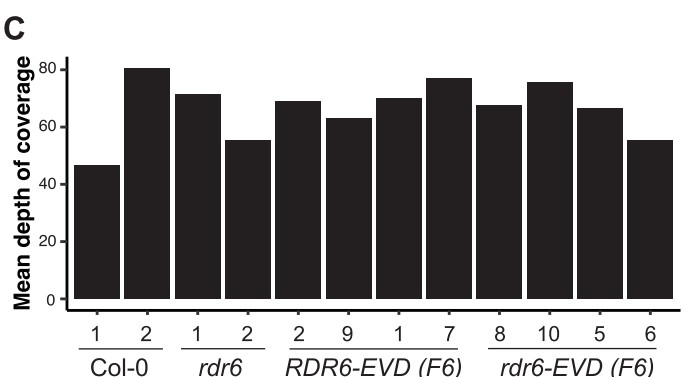

# D

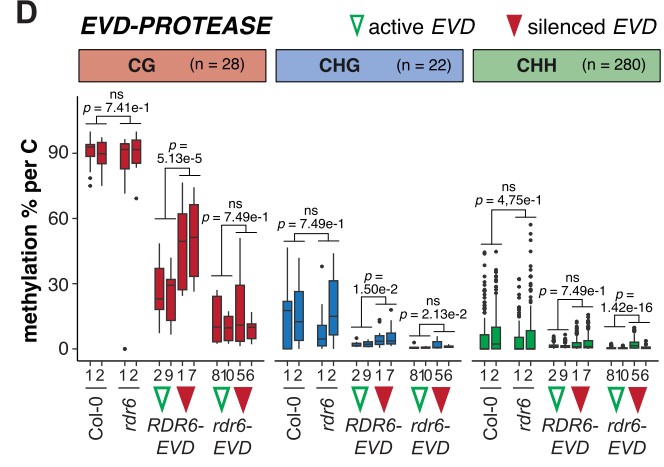

# E

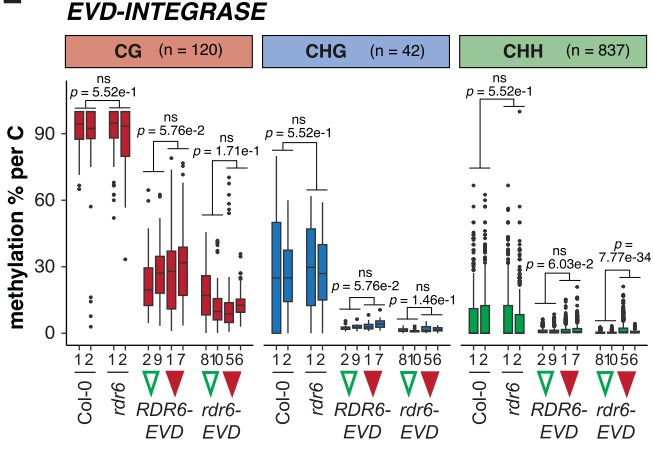

# F

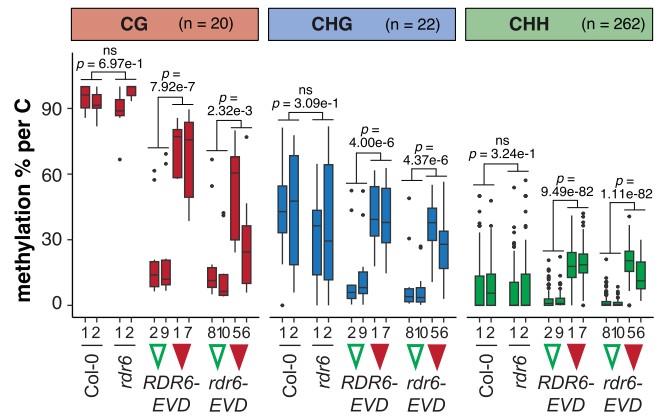

◄ **Figure EV2.  EM-seq libraries general stats and *EVD POL* and *3'LTR* methylation in *RDR6-* and *rdr6-EVD* lines.**

(A) Cytosine-to-thymine conversion rates of the unmethylated chloroplastic DNA for each EM-seq library (see materials and methods for further information). (B) Number of paired-end fragments obtained in each EM-seq library. (C) *Arabidopsis* genome coverage in each EM-seq library. (D–F) EM-seq analysis of DNA methylation (as % per of methylated cytosines) in CG, CHG, and CHH contexts in WT Col-0, *rdr6* and in *RDR6-* and *rdr6-EVD* F6 individuals with active and silenced *EVD* (marked with empty green and filled red arrowheads, respectively, numbers indicate same individuals as in Fig. 2) in: (D) *EVD-Protease*; (E) *EVD-Integrase,* and (F) *EVD-5'LTR*. In panels (D–F), *n* indicates the number of cytosines analyzed for each context per sample. In all boxplots: median is indicated by a solid bar, the boxes extend from the first to the third quartile and whiskers reach to the furthest values within 1.5 times the interquartile range. Dots indicate outliers, as data points outside of the above range. Wilcoxon rank sum test adjusted *p*-values between indicated groups of samples are shown. Differences are considered statistically significant if $p < 0.05$ (5.00e−2) or non-significant (ns) if $p \geq 0.05$. Source data are available online for this figure.

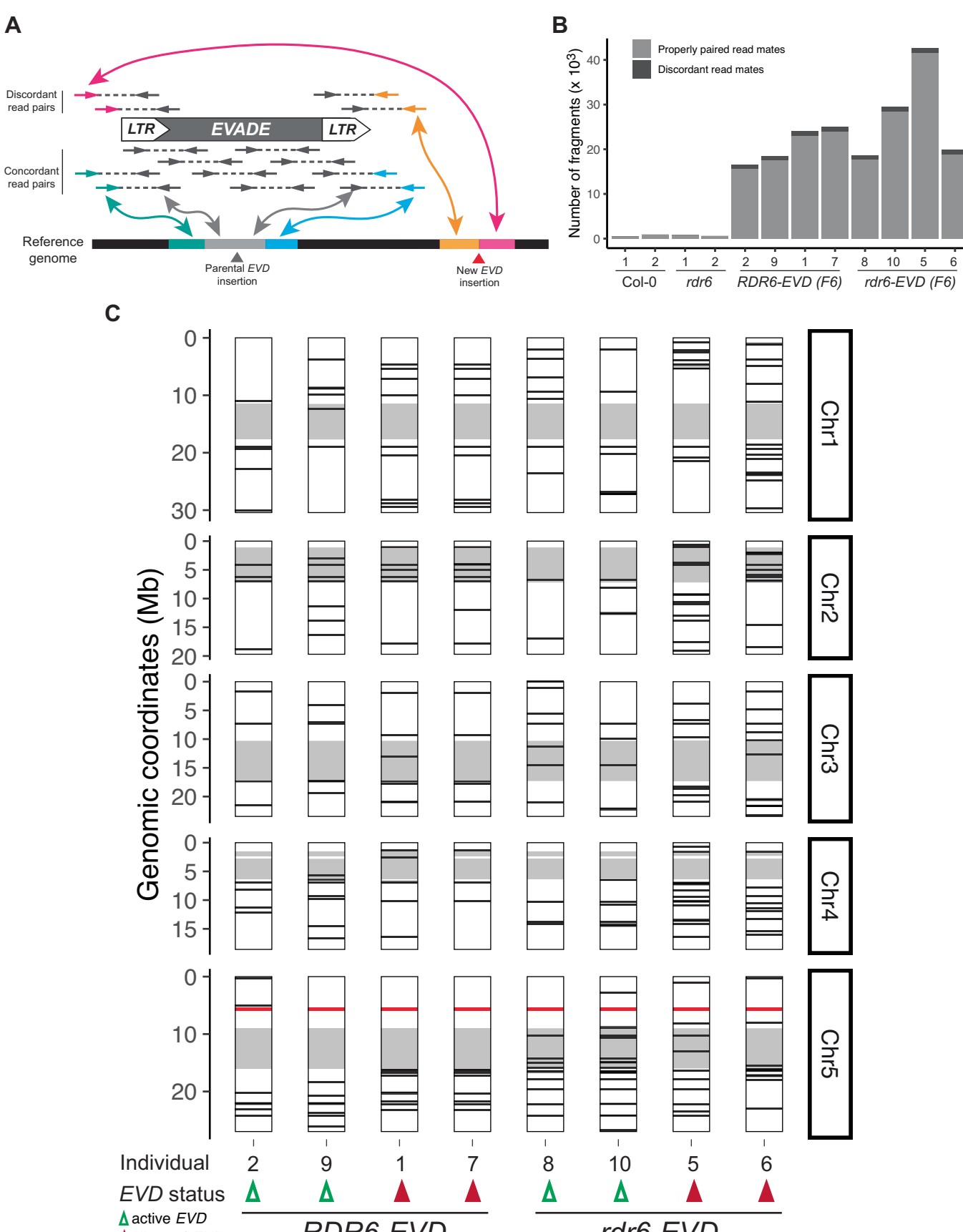

◀ **Figure EV3. Mapping of new *EVD* insertions in *RDR6*- and *rdr6-EVD* lines from EM-seq data.**

(A) Schematic representation of the strategy used to map new *EVD* insertions using discordant read mates from EM-seq. (B) Number of fragments from concordant (properly paired) and discordant read mates mapping to EVD in each of the EM-seq libraries. (C) Genomic location of new EVD insertions mapped through discordant read pair mates in EM-seq data in the *Arabidopsis* genome. Parental EVD (AT5G17125) location is indicated with a red line. New EVD insertions are marked with black lines. Pericentromeric regions in each of *Arabidopsis* five chromosomes are marked in gray. See table provided in corresponding source data for precise chromosome coordinates of each new insertion. Source data are available online for this figure.

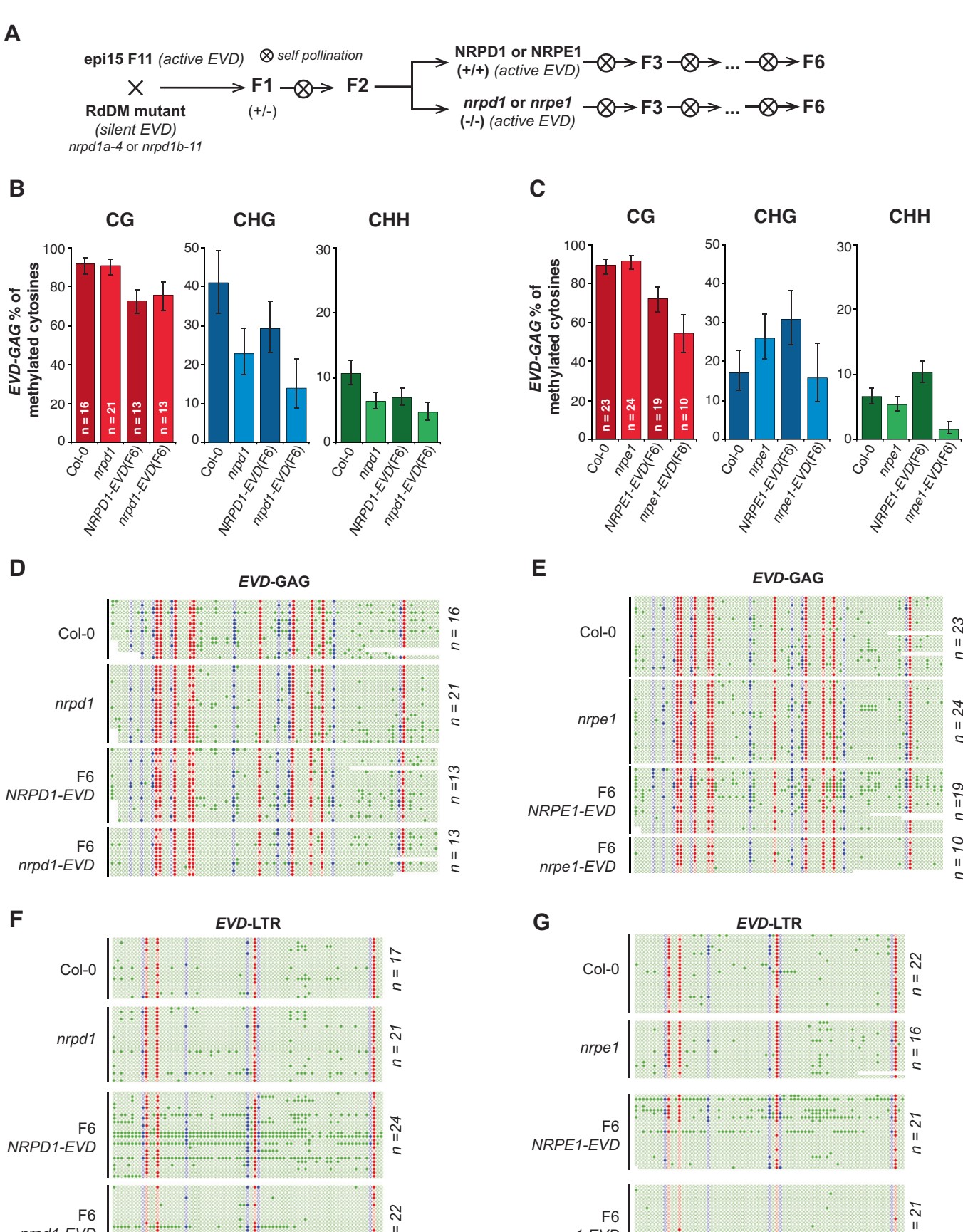

**Figure EV4. BS-PCR analysis of EVD-GAG DNA methylation levels in RdDM mutants.**

(A) Crossing scheme to generate nrpd1- and nrpe1-EVD lines. F2 plants were genotyped to select homozygous WT and mutant lines for each background and propagated through selfing until the F6 generation. (B, C) % methylated cytosines by bisulfite -PCR DNA methylation analysis at EVD-GAG sequences in the F6 generation of *NRPD1-EVD* (B) and *NRPE1-EVD* (C) lines, in both WT (darker shade) and mutant (lighter shade) backgrounds. Col-0, *nrpd1* and *nrpe1* were used as controls. Error bars represent 95% confidence Wilson score intervals of the % of methylated cytosines (C) in each context (CG, CHG, CHH). (D, E) Dot-plot representation of bisulfite-PCR sequencing data for the F6 generation of *NRPD1-EVD* (D) and *NRPE1-EVD* (E) lines, in both WT and mutant backgrounds, at *EVD-GAG* sequences. Col-0 and *nrpd1* or *nrpe1* were used as control for the endogenous parental *EVD* copies. (F, G) Dot-plot representation of bisulfite-PCR sequencing data for the F6 generation of *NRPD1-EVD* (F) and *NRPE1-EVD* (G) lines, in both WT and mutant backgrounds, at *EVD-LTR* sequences. Col-0 and *nrpd1* or *nrpe1* were used as control for the endogenous parental *EVD* copies. In all dot-plots, filled circle represent methylated, empty circles unmethylated cytosines in the CG (red), CHG (blue), and CHH (green) context. Source data are available online for this figure.

| INSERTION ORIGIN | | STRUCTURE | HAIRPIN | OCCURENCE | | | | | | | |
|---|---|---|---|---|---|---|---|---|---|---|---|
| | | | | *RDR6-EVD* (F6) | | | | *rdr6-EVD* (F6) | | | |
| Tandem | Nested | | | 2 | 9 | 1 | 7 | 8 | 10 | 5 | 6 |
| Tail-to-Tail<br>3'-to-3' *LTR* | in Antisense<br>5'-to-3' *LTR* | LTR ><br>< ЯTⅬ | YES | N | N | N | N | N | N | N | N |
| Head-to-Head<br>5'-to-5' *LTR* | in Antisense<br>5'-to-3' *LTR* | < ЯTⅬ<br>LTR > | YES | N | N | N | N | N | N | N | N |
| Head-to-Tail<br>3'-to-5' *LTR* | in Sense<br>5'-to-5' *LTR*<br>3'-to-3' *LTR* | LTR ><br>LTR > | NO | N | N | N | N | N | N | N | N |

**Figure EV5. Summary table for the search of close proximity sense-antisense *EVD-LTR* events.**

Putative origin of *LTR* hairpins as consequence of *EVD* transposition in tandem or nested configurations next to the scheme of the strategy used to find them using discordant read mates from EM-seq where both read mates map to *EVD-LTR*. The table indicates the occurrence and number of read mates found in each sample. Arbitrary threshold of at least three paired read mates was set to confidentially identify two *LTR*s in close proximity. N: non-occurrence, y: presence of at least one discordant paired read mate, Y: presence of discordant paired read mates and above threshold for confident calling.

