## [Peer Review File · EMBO Reports]

PTGS is dispensable for initiation of epigenetic silencing of an active transposon in Arabidopsis

Marieke Trasser, Gregoire Bohl Viallefond, Verónica Barragán-Borrero, Laura Diezma-Navas, Lukas Loncsek, Magnus Nordborg, and Arturo Mari-Ordóñez

Corresponding author(s): Arturo Mari-Ordóñez (arturo.mari-ordonez@gmi.oeaw.ac.at)

Review Timeline:

Submission Date:	27th May 24
Editorial Decision:	1st Jul 24
Revision Received:	12th Sep 24
Editorial Decision:	8th Oct 24
Revision Received:	9th Oct 24
Accepted:	21st Oct 24

Editor: Esther Schnapp

Transaction Report:

Dear Dr. Mari-Ordóñez,

Thank you for the submission of your manuscript to EMBO reports. We have now received the full set of referee reports that is pasted below.

As you will see, all referees acknowledge that the findings are interesting. They only have a few suggestions for how the study could be further improved, and I think all should be addressed. Please let me know in case you disagree and we can discuss the exact revision requirements further, also in a video chat, if you wish.

I would thus like to invite you to revise your manuscript with the understanding that the referee concerns must be fully addressed and their suggestions taken on board. Please address all referee concerns in a complete point-by-point response. Acceptance of the manuscript will depend on a positive outcome of a second round of review. It is EMBO reports policy to allow a single round of major revision only and acceptance or rejection of the manuscript will therefore depend on the completeness of your responses included in the next, final version of the manuscript.

We realize that it is difficult to revise to a specific deadline. In the interest of protecting the conceptual advance provided by the work, we recommend a revision within 3 months (1st Oct 2024). Please discuss the revision progress ahead of this time with the editor if you require more time to complete the revisions.

- 1) A data availability section providing access to data deposited in public databases is missing. If you have not deposited any data, please add a sentence to the data availability section that explains that.
- 2) Your manuscript contains statistics and error bars based on $n=2$. Please use scatter blots in these cases. No statistics should be calculated if $n=2$.

5) a complete author checklist, which you can download from our author guidelines <<https://www.embopress.org/page/journal/14693178/authorguide>>. Please insert information in the checklist that is also reflected in the manuscript. The completed author checklist will also be part of the RPF.

6) Please note that all corresponding authors are required to supply an ORCID ID for their name upon submission of a revised manuscript (<<https://orcid.org/>>). Please find instructions on how to link your ORCID ID to your account in our manuscript tracking system in our Author guidelines <<https://www.embopress.org/page/journal/14693178/authorguide#authorshipguidelines>>

7) Before submitting your revision, primary datasets produced in this study need to be deposited in an appropriate public database (see <https://www.embopress.org/page/journal/14693178/authorguide#datadeposition>). Please remember to provide a reviewer password if the datasets are not yet public. The accession numbers and database should be listed in a formal "Data Availability" section placed after Materials & Method (see also <https://www.embopress.org/page/journal/14693178/authorguide#datadeposition>). Please note that the Data Availability Section is restricted to new primary data that are part of this study. * Note - All links should resolve to a page where the data can be accessed. *
If your study has not produced novel datasets, please mention this fact in the Data Availability Section.

12) All Materials and Methods need to be described in the main text using our 'Structured Methods' format, which is required for all research articles. According to this format, the Methods section includes a Reagents and Tools Table (listing key reagents, experimental models, software and relevant equipment and including their sources and relevant identifiers) followed by a Methods and Protocols section describing the methods using a step-by-step protocol format. The aim is to facilitate adoption of the methodologies across labs. More information on how to adhere to this format as well as a downloadable template (.docx) for the Reagents and Tools Table can be found in our author guidelines: <https://www.embopress.org/page/journal/14693178/authorguide#structuredmethods>.

An example of a Method paper with Structured Methods can be found here: <https://www.embopress.org/doi/full/10.1038/s44320-024-00037-6#sec-4>

I look forward to seeing a revised form of your manuscript when it is ready.

Referee #1:

As of now, still little is known about the initiation of silencing of novel transposable element insertions in the genome. In this manuscript, Trasser et al address this question particularly centering around the role of PTGS in this process. Indeed, PTGS has long been thought to be the initial signal leading to the acquisition of TGS. In this interesting and well-written manuscript, the authors demonstrate that RDR6, a key element of the PTGS pathway, is not required for the establishment of TGS. This is unexpected suggesting that another mechanism exists that can allow a direct transition to TGS without the PTGS intermediary. The authors also show that DNA methylation at the LTR occurs independently of methylation at the GAG sequence.

Overall, this is a very interesting manuscript providing a clear and straight forward story with clear figures and a good discussion.

In the discussion, the authors could eventually speculate a bit more about why the LTRs are specifically targeted by TGS (and not for instance GAG). What makes those targets so special? Is it their copy number (2 time the actual TE copy number)? Is it because they are at the ends of the sequence and hairpin transcripts can result from head-to-head or tail-to-tail EVD insertions?

An important point that I would like to suggest to test is, if the DNA methylation level at the 5' LTR correlates with DNA methylation level at the 3' LTR of individual EVD insertions or whether the two LTRs are independent entities from a silencing point of view. This can readily be tested with the nice DNA methylation data that was produced for the inserted EVD copies.

It is important that the authors note that most PCR based approaches used in this manuscript will capture integrated TEs but also extrachromosomal copies thereof. This could for instance bias some of the DNA methylation results. I suggest to add some additional notes of cautions about this in the manuscript.

Minor comments:

Lines 178-179 a part of the sentence is duplicated.

Line 358: Please introduce RPP4 solo_LTR to the reader.

Referee #3:

This ms addresses the chicken and egg question of how TEs get transcriptionally silenced in first place before TGS is maintained by the classical PolIV-RdDM pathway. The authors use epigenetically reactivated EVD as a model system. They previously showed that EVD produce RNAs that are used by RDR6 to produce dsRNA that are diced by DCL4 into 21-nt siRNAs, which guide DNA methylation in EVD coding sequence. They also showed that this process does not prevent EVD to transpose actively. They proposed that when EVD reach 40-50 copies, DCL4 gets saturated, allowing DCL3 to produce 24-nt that that attract AGO4, PolIV and PolV to initiate TGS in the EVD LTR, which prevents transcription and stops transposition (Mari-Ordonez et al, 2013). A refined model of TE silencing was proposed by Sigman et al, 2021, in which any size of siRNAs can be directly loaded into AGO4 to initiate TGS, allowing the possibility that primary siRNAs are made in different ways and not only through the action of RDR6.

Here the authors address whether RDR6 is absolutely required to initiate EVD TGS. They show that EVD can become silenced

by TGS in plants lacking RDR6. However, unlike RDR6-proficient plants, which trigger EVD TGS in a dosage-dependent manner (above 40-50 copies), RDR6-deficient plants trigger EVD TGS in a stochastic manner, suggesting that the underlying mechanism is not dosage-dependent but more likely locus-dependent, i.e. TGS is initiated when one of the de novo created EVD locus produce 24-nt siRNAs in a RDR6-independent manner, which trigger TGS. They propose several hypotheses to explain how TGS could be triggered in a RDR6-independent manner, including the possibility that DNA breaks attract the methylation machinery, or that a heterochromatin state at the site of insertion spreads into the LTR, or that insertion as an inverted repeat produce dsRNA without the requirement of RDR6, which are diced into 24-nt by DCL3.

The data are solid, but I think the authors could go further in exploiting these data. Because they have all the EVD insertion sites in *rdr6* plants that have either silenced EVD copies or non-silenced EVD copies, they should look for common features present in all silenced lines and absent in all non-silenced lines, i.e. presence of inverted repeat, particular DNA methylation or chromatin state surrounding the insertion, etc.

In my opinion this ms deserves publication, but I think the authors should insist on the fact that RDR6 allows establishing TGS in a very reliable manner (no plants carrying more than 40-50 copies can be obtained), and that at least one alternative way exist, but that it is less reliable because it requires a particular type of insertion (which probability to arise, of course, increases as the copy number increases).

Referee #4:

In this manuscript, Trasser et al. aim at testing whether post-transcriptional gene silencing (PTGS) is required for the de novo establishment of epigenetic silencing at an active transposable element (TE) in *Arabidopsis thaliana*. To this end, they introduce an active LTR-retroelement (EVD) that is not targeted neither by PTGS nor by TGS in a mutant background (*rdr6*) where PTGS is impaired. By tracking EVD activity through generations in terms of transcription, methylation state, sRNAs, and copy-number in both the *rdr6* mutant and their uncompromised siblings, they find clear evidence that, although PTGS appears to facilitate TGS, it is not required for de novo establishment of silencing of EVD. Using a similar strategy, the authors then follow EVD activity in Pol-IV and Pol-V mutants, and show that both are required for TGS, thus demonstrating that initiation of silencing requires the canonical RNA-directed DNA methylation (RdDM). In contrast, RDR6-RdDM, which does not rely on Pol-IV, is not sufficient for full EVD silencing although it appears to contribute some level of DNA methylation deposition over both EVD internal and LTR domains.

By challenging the established model for the initiation of silencing at active TE sequences through an elegant genetic approach, the authors bring an important new set of evidence to the age-old conundrum of how cells recognize active TEs for entirely de novo silencing. In this light, the authors discuss several alternative mechanisms through which the silencing may be triggered, notably the possibility that integration into sites of pre-existing heterochromatin domains may subject one or few EVD copies to RdDM, similar in a way to the mechanism proposed in *Drosophila* for piRNA clusters. In this regard, the data they generate, through a combined EM-seq split-read approach, about the location and methylation state of EVD insertions in a panel of silenced and active *rdr6* lines, may hold more insights than the simple analysis of pericentromeric enrichments performed, especially as pericentromeres are not particularly strong RdDM targets. Comparing instead insertion sites with established RdDM targets may help identify candidate TE insertions responsible for the initiation of RdDM, although the experimental confirmation of causality would be indeed beyond the scope of the present study.

minor points

l. 55-56: Who knows what is the end goal of TE silencing? What is clear however is that one consequence is indeed a limitation (TEs still manage to transpose fine though!) of their mobility. Rephrase beginning of sentence accordingly.

l. 168: typo at mutant

l. 182: F6 should be represented by biological replicates in Fig. 1D or in Fig. S1 in order to distinguish the amount of the large variation observed that is due to biological rather than technical origin.

l. 432: typo at shown

Dear Editor,

Please find below our point-by-point response to reviewer's. We thank the referees for their assessment and appreciation of our work and constructive suggestions to improve the quality of the manuscript. We have addressed all the referees' comments and performed some analysis and corrections to the text as requested and or suggested. Resulting changes in the text are **highlighted in yellow** in the revised manuscript but also indicated here for convenience.

Referee #1:

As of now, still little is known about the initiation of silencing of novel transposable element insertions in the genome. In this manuscript, Trasser et al address this question particularly centering around the role of PTGS in this process. Indeed, PTGS has long been thought to be the initial signal leading to the acquisition of TGS. In this interesting and well-written manuscript, the authors demonstrate that RDR6, a key element of the PTGS pathway, is not required for the establishment of TGS. This is unexpected suggesting that another mechanism exists that can allow a direct transition to TGS without the PTGS intermediary. The authors also show that DNA methylation at the LTR occurs independently of methylation at the GAG sequence.

Overall, this is a very interesting manuscript providing a clear and straight forward story with clear figures and a good discussion.

1) In the discussion, the authors could eventually speculate a bit more about why the LTRs are specifically targeted by TGS (and not for instance GAG). What makes those targets so special? Is it their copy number (2 time the actual TE copy number)? Is it because they are at the ends of the sequence and hairpin transcripts can result from head-to-head or tail-to-tail EVD insertions?

We thank the referee for this suggestion. We have indeed neglected this important point in the discussion as the presence of repeated sequences (LTRs), flanking TE coding regions, makes LTR-RTEs display a structure that might favor their targeting by TGS.

To address this important point, we have now added a paragraph in the discussion (starting at line 695 in the revised manuscript) to take this point into consideration and speculate about the role that LTRs might play in EVD silencing in the contexts proposed by the referee. Additionally, while reviewing the literature thanks to the referee suggestion, we found that presence/absence of sequence duplications flanking transgenic reporters (similar to LTR-RTE sequence configurations) have a strong impact in its epigenetic features. We have also discussed that literature in the context of TE silencing in the new paragraph.

We believe this new paragraph has significantly improved our discussion and we hope it has addressed the referee's suggestion.

2) An important point that I would like to suggest to test is, if the DNA methylation level at the 5' LTR correlates with DNA methylation level at the 3' LTR of individual EVD insertions or whether the two LTRs are independent entities from a silencing point of view. This can readily be tested with the nice DNA methylation data that was produced for the inserted EVD copies.

We are grateful to the referee for raising this important point and we have performed the test as suggested. The analysis, now shown in Figure 5B in the revised manuscript, has revealed an important consequence of the loss of PTGS on EVD silencing. In addition to the increased heterogeneity in LTR methylation levels in absence of RDR6 following the switch to TGS as previously shown (former Figure 4E, now Figure 5B), the methylation levels between LTRs of the same insertion also show higher variability than in the PTGS-competent lines. While in RDR-EVD lines CG methylation levels of silenced insertions are generally above 50% in both LTRs and tend to display a positive correlation, in the rdr6-EVD lines, such correlation is lower as several insertions display high methylation levels in one LTR but not the other (without any preference for 5' or 3' LTR).

This analysis is now part of Figure 5 (Figure 5B) and of the results (starting at line 353) in the revised manuscript. The finding is also mentioned in the discussion (line 565) and in the conclusion paragraph (line 728). We thank again the referee as the analysis suggested, and the results obtained, have made a significant contribution to improve the manuscript and we hope its incorporation into the revised version satisfactorily addresses the referee's point.

3) It is important that the authors note that most PCR based approaches used in this manuscript will capture integrated TEs but also extrachromosomal copies thereof. This could for instance bias some of the DNA methylation results. I suggest to add some additional notes of cautions about this in the manuscript.

The referee is correct at pointing this out, indeed we took the precaution of considering such issue to explain the discrepancy between copy numbers measured by qPCR and EM-seq as PCR-based methods will capture both integrated and extra-chromosomal (ecDNA) EVD copies in tissues/samples with active EVD (lines 333 to 338 in revised manuscript). However, we did not consider that the same technical bias applies to our DNA methylation levels by BS-PCR. Although EVD is only expressed in some cells of the adaxial L2 cell layer, and the contribution of ecDNA should dilute within the rest of the tissue, the differences in methylation reported here might be affected by the technical limitation pointed out by the referee. Therefore, to account for such biases, make the reader aware of them and be more precise in the manuscript, we have:

- Through the text eliminated any reference to methylation of EVD insertions and simply refer it to EVD methylation whenever it was estimated using the BS-PCR method.
- When the BS-PCR method for quantification of EVD methylation is introduced in the results section, we have added a statement to reflect and introduce to the reader such technical bias. Starting at line 406 in the revised manuscript the text now reads (introduced text is underlined and deleted is strikethrough): “Methylation of EVD ~~insertions~~ was assessed in F6 bulks by Sanger sequencing of PCR amplicons from bisulfite-treated DNA (BS-PCR). Although this PCR-based method does not discern between integrated and extra-chromosomal integration intermediates in lines with active EVD, it allowed us to estimate overall EVD methylation levels.”
- An additional caution note has been added to the discussion section where the results obtained through BS-PCR are discussed (with regards to the potential role of RDR6-RdDM in spreading methylation from GAG to LTR). From line 432 in the revised manuscript, it now reads as (introduced text is underlined): “...promote DNA methylation spreading into the nearby LTR. Still, as the BS-PCR used here amplifies both integrated and ecDNA EVD copies, a similar EM-seq strategy as used for the EVD-rdr6 F6 individuals will be required in the future to obtain LTR methylation levels of individual insertions in nrp1- and nrpe1-EVD lines.”

We are hopeful that these text additions address the referee’s issues and clarify the limitations of the technique to the reader as suggested by the referee.

4) Minor comments:

4.1) Lines 178-179 a part of the sentence is duplicated.

This has been corrected.

4.2) Line 358: Please introduce RPP4 solo_LTR to the reader.

The RPP4 solo_LTR was previously introduced in the results section, line 293 of the revised manuscript: “...of the EVD-derived solo-LTR present in the promoter of RECOGNITION OF PERONOSPORA PARASITICA 4 (RPP4, AT4G16860), which can be methylated in trans by EVD-LTR 24-nt siRNAs following an EVD de novo silencing event (Mari-Ordóñez et al. 2013).” However, we did not establish the nomenclature of RPP4 solo_LTR as pointed out by the referee. To introduce the RPP4 solo_LTR to the reader, the text has been modified in the revised version of the manuscript as follows from line 293 (introduced text is underlined): “...methylation status of the EVD-derived solo-LTR present in the promoter of RECOGNITION OF PERONOSPORA PARASITICA 4 (RPP4, AT4G16860), referred to as RPP4 solo_LTR hereafter,...”

Furthermore, to reinforce the connection to EVD and remind the reader about its methylation dependency on EVD switch to TGS, we have modified the text as follows from line 372 in the revised manuscript (introduced text is underlined): “...RdDM for less generations. This might be the case especially in the rdr6-EVD #6 line, where RPP4 solo_LTR methylation levels, which depends on EVD-LTR-derived 24-nt siRNAs, are also lower than in the other EVD-silenced lines (Fig. 3D)”.

Referee #3:

This ms addresses the chicken and egg question of how TEs get transcriptionally silenced in first place before TGS is maintained by the classical PolIV-RdDM pathway. The authors use epigenetically reactivated EVD as a model system. They previously showed that EVD produce RNAs that are used by RDR6 to produce dsRNA that are diced by DCL4 into 21-nt siRNAs, which guide DNA methylation in EVD coding sequence. They also showed that this process does not prevent EVD to transpose actively. They proposed that when EVD reach 40-50 copies, DCL4 gets saturated, allowing DCL3 to produce 24-nt that that attract AGO4, PolIV and PolV to initiate TGS in the EVD LTR, which prevents transcription and stops transposition (Mari-Ordóñez et al, 2013). A refined model of TE silencing was proposed by Sigman et al, 2021, in which any size of siRNAs can be directly loaded into AGO4 to initiate TGS, allowing the possibility that primary siRNAs are made in different ways and not only through the action of RDR6.

Here the authors address whether RDR6 is absolutely required to initiate EVD TGS. They show that EVD can become silenced by TGS in plants lacking RDR6. However, unlike RDR6-proficient plants, which trigger EVD TGS in a dosage-dependent manner (above 40-50 copies), RDR6-deficient plants trigger EVD TGS in a stochastic manner, suggesting that the underlying mechanism is not dosage-dependent but more likely locus-dependent, i.e. TGS is initiated when one of the de novo created EVD locus produce 24-nt siRNAs in a RDR6-independent manner, which trigger TGS. They propose several hypotheses to explain how TGS could be triggered in a RDR6-independent manner, including the possibility that DNA breaks attract the methylation machinery, or that a heterochromatin state at the site of insertion spreads into the LTR, or that insertion as an inverted repeat produce dsRNA without the requirement of RDR6, which are diced into 24-nt by DCL3.

1) The data are solid, but I think the authors could go further in exploiting these data. Because they have all the EVD insertion sites in *rdr6* plants that have either silenced EVD copies or non-silenced EVD copies, they should look for common features present in all silenced lines and absent in all non-silenced lines, i.e. presence of inverted repeat, particular DNA methylation or chromatin state surrounding the insertion, etc.

*We thank the referee from making this suggestion. We had previously considered such analysis although we were concerned by the small sample size, the *met1-epiRIL* origin of our lines and the technical limitations of using EM-seq for some of those analysis instead of long read based genome sequencing approaches. As similar analysis have also been requested by referee #4 we have performed the following analysis, which have been now incorporated in the manuscript in two new sub-sections in the results at lines 441 and 523 and two additional figures (Figure 7 ad Figure EV5) in the revised manuscript:*

- *Search for EVD integration configurations potentially leading to LTR hairpins: Using a similar approach to map new insertions, we have used discordant paired read mates to look for LTR hairpins by searching for read pairs in which both mates map to EVD-LTR but in antisense orientation to each other (Figure EV5). We did not find such a case in any of the samples (Figure EV5). However, this approach is technically limited as to detect LTR in antisense orientation to each other they have to closer than the insert range of the EM-seq library (300-700 bp). Hence, the analysis was expanded to search for antisense EVD nested insertions, which can potentially also result in hairpins or dsRNA through convergent transcription. To do so, we searched for mate pairs in which one read mapped to an LTR and the other to EVD sequences in antisense orientation (Figure 7A). With this approach several samples contained at least one mate paired in such configuration. However, as those can also be the result of sequencing errors, we set as a confidence threshold that at least two mate pairs were found by LTR per location to allow location of the nested EVD insertion from both ends of the nested insertion and identify the target site duplication (TDS) as signature of a bona-fide EVD transposition event. Only one *rdr6*-EVD individual, which happens to have silenced EVD, fulfills such requirements (Figure 7A). The structure on such nested insertion can potentially lead to the generation of siRNAs if transcribed as depicted in Figure 7B, although it will require future investigation to know if such events can lead to siRNAs and TGS. These results are depicted in the revised version of the manuscript from line 447 to 476. A comment to such results is also now present in the discussion (line 664 to 668)*
- *As only one of the EVD silenced *rdr6*-EVD individuals contained an antisense nested EVD insertion, we searched for alternative triggers of silencing. As suggested by the referees, we investigated the epigenetic status of the landing sites of new EVD copies as transposition into an RdDM locus might suffice to initiate silencing. Given that active EVD copies were introgressed into *rdr6* by crossing with the *met1*-derived *epiRIL*#15, we were concerned that the epigenome of the lines used in this study might differ from that of WT *Arabidopsis* (Reinders et al. 2009). Hence, using our EM-seq dataset, the DNA methylation status of EVD landing sites was inferred from the DNA methylation patterns of other individuals without novel EVD insertions at the same positions as explained in the revised version (line 476 to 516 and Figure 7C). The analysis shows that while most new EVD insertions happen in unmethylated loci, a few do in CG methylated and even in locations with CHG and CHH methylation (Figure 7D). However, insertions in positions containing non-CG methylation are found in individuals with either active or silenced EVD (Figure 7D). As the pathway (RdDM or CMT) dependency of non-CG methylation is difficult to infer from DNA methylation patterns, we used the data from (Sasaki et al. 2022), where the DNA methylated regions of the *Arabidopsis* genome were sorted according to their genetic dependency based on a previous study of the methylomes of several *Arabidopsis* mutants by the Jacobsen's lab (Stroud et al. 2013). Again, given the *epiRIL* origin of one of the parental lines, such genetic dependencies might not hold true in our lines and this has been made clear to the reader in the text (line 506 to 509). Nonetheless, despite the caveats, the analysis indicates that one of the *rdr6*-EVD lines with silenced EVD (the one without the nested inverted insertion) carries two insertions in potential RdDM loci that might have contributed to initiate TGS. However, one of the lines with active EVD also carries an insertion in a predicted RdDM locus. Therefore, we cannot take any strong conclusion of such analysis, but it shows that insertions into regions with non-CG methylation can take place, which we only speculated about before, although is unclear if they suffice to initiate TGS in all new insertions as now shown in the discussion (line 676 to 685).*

- Lastly, as we had the data from the previous analysis, we investigated if EVD transposition caused any changes in local DNA methylation at landing sites upon insertion or silencing. As DNA methylation might increase in response to transposition or DNA damage caused by it, we compared EVD insertion sites methylomes with/without EVD, before/after silencing. The results of this analysis, now in results (line 523 to 536 and Figure 7E) show that no major changes occur in response to transposition. We only observed a very localized increase in CHG and CHH around the insertion site after EVD silencing with very limited spreading (Figure 7E). These new results have also been reflected in the discussion (lines 648-649).

We hope these new analyses address the referee's comment. Although they do not provide a definitive answer to the underlying cause behind the initiation of TGS in absence of RDR6-RdDM, they provide evidence that some of the potential causes previously speculated in the discussion do happen during an EVD transposition burst. Because of that, we believe these new analysis in response to the referee's suggestions have greatly improved the quality of the manuscript and open further research directions.

2) In my opinion this ms deserves publication, but I think the authors should insist on the fact that RDR6 allows establishing TGS in a very reliable manner (no plants carrying more than 40-50 copies can be obtained), and that at least one alternative way exist, but that it is less reliable because it requires a particular type of insertion (which probability to arise, of course, increases as the copy number increases).

We agree with the referee that the of robustness of TGS installation provided by RDR6, as opposed to the stochasticity observed in its absence, which was shortly addressed in the discussion, has not been properly highlighted throughout the text. To correct this, we have made the following modifications through the revised manuscript to ensure this fact receives enough attention:

- *Abstract:* we have added the following sentence (at line 43): "However, in contrast to plants with RDR6, where TGS installation robustly happens at a threshold of 40-50 copies, in its absence, silencing takes place more stochastically."
- *Results:* In the subsection where we show the lack of correlation between TGS installation and EVD copy number in the *rdr6* background. Starting at line 244 in the revised manuscript the text now reads (introduced text is underlined): "Consequently, while PTGS seems to facilitate the establishment of TGS in a reliable manner, in the absence of RDR6 activity, no clear copy number threshold for TGS installation was observed."
- *Discussion:* When discussing about how PTGS might facilitate the installation of TGS, starting at line 565 in the revised manuscript the text now reads (introduced text is underlined): Nonetheless, given the consistent installation of silencing in presence of PTGS, limiting EVD proliferation beyond 40-50 copies, ..."
- *Conclusion:* The conclusion/summary paragraph of the discussion, starting at line 725 of the revised manuscript, has also been modified to stress this point and it now reads (introduced text is underlined): "In conclusion, our study reveals that, while PTGS activity contributes to limit TE proliferation ensuring the faithful setting of TGS, alone is insufficient to explain the de novo initiation of epigenetic silencing of an active proliferative TE in the Arabidopsis genome. TGS initiation in absence of PTGS is more stochastic and, although we have not identified the trigger event, our results open further questions about genome defense mechanisms. Several of the above-mentioned phenomena, alone or in combination, could contribute to de novo silencing of EVD and TEs, and deserve future investigation to gain insights into the mechanisms of defense against genomic parasites."

We hope the modifications and clarifications in the manuscript address the referee's valid comment.

Referee #4:

In this manuscript, Trasser et al. aim at testing whether post-transcriptional gene silencing (PTGS) is required for the de novo establishment of epigenetic silencing at an active transposable element (TE) in *Arabidopsis thaliana*. To this end, they introduce an active LTR-retroelement (EVD) that is not targeted neither by PTGS nor by TGS in a mutant background (*rdr6*) where PTGS is impaired. By tracking EVD activity through generations in terms of transcription, methylation state, sRNAs, and copy-number in both the *rdr6* mutant and their uncompromised siblings, they find clear evidence that, although PTGS appears to facilitate TGS, it is not required for de novo establishment of silencing of EVD. Using a similar strategy, the authors then follow EVD activity in *Pol-IV* and *Pol-V* mutants, and show that both are required for TGS, thus demonstrating that initiation of silencing requires the canonical RNA-directed DNA methylation (RdDM). In contrast, RDR6-RdDM, which does not rely on *Pol-IV*, is not sufficient for full EVD silencing although it appears to contribute some level of DNA methylation deposition over both EVD internal and LTR domains.

By challenging the established model for the initiation of silencing at active TE sequences through an elegant genetic approach, the authors bring an important new set of evidence to the age-old conundrum of how cells recognize active TEs for entirely de novo silencing. In this light, the authors discuss several alternative mechanisms through which the silencing may be triggered, notably the possibility that integration into sites of pre-existing heterochromatin domains may subject one or few EVD copies to RdDM, similar in a way to the mechanism proposed in *Drosophila* for piRNA clusters.

1) In this regard, the data they generate, through a combined EM-seq split-read approach, about the location and methylation state of EVD insertions in a panel of silenced and active *rdr6* lines, may hold more insights than the simple analysis of pericentromeric enrichments performed, especially as pericentromeres are not particularly strong RdDM targets. Comparing insertion sites with established RdDM targets may help identify candidate TE insertions responsible for the initiation of RdDM, although the experimental confirmation of causality would be indeed beyond the scope of the present study.

We are thankful to the referee this suggestion. As similar comment has been made by referee #3, we have now performed further analysis to investigate:

- *EVD insertions nearby or within itself that can potentially lead to the generation of RNA hairpins or other small RNA sources.*
- *The DNA methylation status of new EVD insertion sites before transposition and their dependency on RdDM.*
- *Changes in DNA methylation at those sites with/without EVD before/after silencing.*

*A detailed explanation of those analyses and the results obtained has been developed in response to referee's #3 comment #1 and can be found in the new manuscript version (lines 441 to 536), throughout the discussion and in the new Figure 7 and Figure EV5. In short, although no changes in local DNA methylation have been found upon EVD insertion, both antisense nested EVD insertions, potentially giving rise to hairpins, and insertions within putative RdDM loci have been found within *rdr6*-EVD silenced lines. Given the small sample size of our study, these results do not suffice to clearly determine whether they are a common trigger of TGS in absence of RDR6-RdDM. Moreover, further analysis using appropriate sequencing methods and other techniques will be needed to investigate if they can initiate silencing. Nonetheless, the fact that we have found them within the EVD lines is a strong suggestion that they can occur during TE colonization events as speculated in our discussion and rightfully suggested by the referee. Hence, we really appreciate the suggestion as the analysis has significantly improved the manuscript, and we hope they have addressed the referee's comment.*

2) minor points:

2.1) l. 55-56: Who knows what is the end goal of TE silencing? What is clear however is that one consequence is indeed a limitation (TEs still manage to transpose fine though!) of their mobility. Rephrase beginning of sentence accordingly.

Indeed. The sentence has been rephrased to not assume any end goal of TE silencing as suggested by the referee. Starting at line 55 in the revised manuscript the text now reads (introduced text is underlined and deleted is strikethrough): "~~To prevent their activity,~~ TEs are mostly transcriptionally repressed across genomes through the action of epigenetic silencing mechanisms, limiting TE activity."

2.2) l. 168: typo at mutant.

This has been corrected.

2.3) l. 182: F6 should be represented by biological replicates in Fig. 1D or in Fig. S1 in order to distinguish the amount of the large variation observed that is due to biological rather than technical origin.

For all qPCR data, the data is now plotted as suggested by the referee. Technical replicates are shown for individual biological samples and bars displaying technical error removed (in agreement with the journal policy). When more than 2 biological samples were used for estimating the mean, the independent values for each biological replicate are also displayed:

- *In Figure 1C, 1D and EV1 (former S1), the EVD copy number and expression levels across generations in the two independent populations are shown separately, and not averaged as biological replicates, together with the individual datapoints from technical replicates.*
- *Similarly, in Figure 2, qPCR data for EVD copy number and expression in individual F6 plants displays technical replicate data points.*

- *In Figure 6A and 6B (former Figure 5A and 5B) individual values for each biological replicate are shown.*

2.4) I. 432: typo at shown

This has been corrected.

REFERENCES:

Reinders J, Wulff BBH, Mirouze M, Ordóñez AM, Dapp M, Rozhon W, Bucher E, Theiler G, Paszkowski J. 2009. Compromised stability of DNA methylation and transposon immobilization in mosaic Arabidopsis epigenomes. *Genes & development* **23**: 939–950.

Sasaki E, Gunis J, Reichardt-Gomez I, Nizhynska V, Nordborg M. 2022. Conditional GWAS of non-CG transposon methylation in Arabidopsis thaliana reveals major polymorphisms in five genes. *PLoS Genet* **18**: e1010345.

Stroud H, Greenberg MVC, Feng S, Bernatavichute YV, Jacobsen SE. 2013. Comprehensive Analysis of Silencing Mutants Reveals Complex Regulation of the Arabidopsis Methylome. *Cell* **152**: 352–364.

Dear Dr. Marí-Ordóñez,

Thank you for the submission of your revised manuscript. We have now received the enclosed reports from the referees that were asked to assess it, and I am happy to say that all support its publication now.

Only a few editorial requests will need to be addressed before we can proceed with the official acceptance of your manuscript:

- Please correct "Data and code availability" to "Data Availability Section"
- Please correct "Conflict of interests" to "Disclosure Statement and Competing Interests"
- Please remove the author credits from the ms file. All credits need to be entered during ms submission.
- Please answer the question about blinding in the author checklist (statistic section).
- Each funder that is acknowledged in the ms file also needs to be entered during online ms submission.
- Supplementary Figure 8 is not a correct callout, please correct.
- The APPENDIX file "List and sequences of oligonucleotides used in this study" - could be better renamed to Table EV1. Please update the ms callouts accordingly. Alternatively, the oligo sequences can be added to the Reagent and Tools table.
- The Reagents and Tools table is only needed as a separate file (not part of the methods section).
- The source data for the EV figures need to be grouped into one folder.
- EV figure legend titles in the ms need correction: it should be Figure EV1, etc. instead of Expanded View Figure 1, etc.
- Please provide the exact p values in the legends of figures 3A-D; 6A, B; EV2 D-F (as reasonable).
- Please note that in figures 2B-C; 3A-D; EV 2D-F there is a mismatch between the annotated p values in the figure legend and the annotated p values in the figure file that should be corrected.
- Please note that information related to n is missing in the legends of figures 3A-D;5A; EV2 D-F.
- Please note that the measure of center for the error bars needs to be defined in the legends of figures 6A, B, E, F; EV4 B, C."

I would like to suggest some minor changes to the abstract that needs to be written in present tense:

Transposable elements (TEs) are repressed in plants through transcriptional gene silencing (TGS) maintained by epigenetic silencing marks such as DNA methylation. However, the mechanisms by which silencing is first installed remains poorly understood in plants. Small interfering (si)RNAs and post-transcriptional gene silencing (PTGS) are believed to mediate the initiation of TGS by guiding the first deposition of DNA methylation. To determine how silencing installation works, we use ÉVADÉ (EVD), an endogenous retroelement in Arabidopsis, able to recapitulate true de novo silencing with a sequence of PTGS followed by TGS. To test whether PTGS is required for TGS, we introduce active EVD into RNA-DEPENDENT-RNA-POLYMERASE-6 (RDR6) mutants, an essential PTGS component. EVD activity and silencing are monitored across several generations. In the absence of PTGS, silencing of EVD is still achieved through installation of RNA-directed DNA methylation (RdDM). Our study shows that PTGS is dispensable for de novo EVD silencing. Although we cannot rule out that PTGS might facilitate TGS, or control TE activity, initiation of epigenetic silencing can take place in its absence.

EMBO press papers are accompanied online by A) a short (1-2 sentences) summary of the findings and their significance, B) 2-3 bullet points highlighting key results and C) a synopsis image that is exactly 550 pixels wide and 200-600 pixels high (the height is variable). The synopsis image should provide a sketch of the major findings, like a graphical abstract. Please note that text needs to be readable at the final size. Please send us this information along with the final manuscript.

Kind regards,
Esther

Referee #3:

I am totally satisfied by the way the authors addressed my concerns as well as the concerns of the other reviewers

Referee #4:

In this revised version of the manuscript, Trasser et al. have satisfyingly corrected the minor revisions suggested and have also added, following reviewers' suggestions, very informative new analyses on the correlation between LTR methylation (Fig. 5B) and on the characteristics of the EVD insertion sites they identify in rdr6 or RDR6 lines that have either silenced or active EVD copies (Fig. 7D). Although this latter analysis does not provide a definitive answer to the underlying cause behind the initiation of TGS in the absence of RDR6-RdDM it adds strong evidence that some of the previously speculated causes of silencing (nested inverted insertions, insertion within pre-existing RdDM target) do occur following EVD transposition which improves significantly the depth of the study.

Apart from a typo in the title of Fig. EV4 (BS-PRC instead of BS-PCR), I have no further comments or suggestions.

All editorial and formatting issues were resolved by the authors.

Dr. Arturo Mari-Ordóñez
Gregor Mendel Institute of Molecular Plant Biology
Dr. Bohr-Gasse 3
Vienna, Vienna 1030
Austria

Dear Arturo,

I am very pleased to accept your manuscript for publication in the next available issue of EMBO reports. Thank you for your contribution to our journal.
